# Prolonged Aggressive Experience Accelerates Resolution of Inflammation in Blood and Microglia After Repeated LPS Treatment

**DOI:** 10.3390/ijms262412007

**Published:** 2025-12-13

**Authors:** Anastasia Mutovina, Anna Sapronova, Kseniya Ayriyants, Yulia Ryabushkina, Julia Khantakova, Polina Mezhevalova, Polina Ritter, Natalya Bondar

**Affiliations:** 1Institute of Cytology and Genetics (ICG), Siberian Branch of Russian Academy of Sciences (SB RAS), Prospekt Akad. Lavrentyeva 10, 630090 Novosibirsk, Russia; sapronovann@gmail.com (A.S.); kaayriyants@bionet.nsc.ru (K.A.); ryabushkina@bionet.nsc.ru (Y.R.); khantakovain@bionet.nsc.ru (J.K.); p.mezhevalova@g.nsu.ru (P.M.); kisaretova@bionet.nsc.ru (P.R.); 2Department of Natural Sciences, Novosibirsk State University, Pirogova Street 2, 630090 Novosibirsk, Russia

**Keywords:** pathological aggression, neuroinflammation, LPS, microglia, dopamine, hypothalamus, nucleus accumbens

## Abstract

This study investigated how prolonged aggression in male CD1 mice alters responses to chronic LPS (lipopolysaccharide)-induced inflammation. Experience of aggression induced pathological aggression in 36% of mice. Following LPS, aggressors resolved systemic inflammation within five days—evidenced by normalized locomotor activity, WBC (white blood cells), and lymphocyte counts—while controls remained inflamed. LPS did not alter established aggression or anxiety. Furthermore, aggressors demonstrated accelerated inflammation resolution in the brain, showing a higher proportion of resting microglia and a lower percentage of activated microglia following LPS-induced inflammation compared to control animals. Gene expression analysis revealed a more pronounced inflammatory response in the hypothalamus than in the nucleus accumbens. Aggressive mice exhibited a profile associated with inflammation resolution, indicated by increased expression of the *Trem2* gene. These differential immune responses may be modulated by the dopaminergic system. Elevated *Drd1* gene expression in the hypothalamus could possibly contribute to the anti-inflammatory signaling, while changes in nucleus accumbens dopaminergic activity, involving D2 receptor activation, appear linked to the development of pathological aggression. Thus, this study demonstrates that prolonged aggression induces persistent changes in behavioral, neuroimmune, and neuroendocrine systems in male CD1 mice. Aggressive animals develop a distinct neuroimmune phenotype characterized by accelerated resolution of both systemic and brain inflammation following LPS challenge.

## 1. Introduction

Aggressive behavior is part of the normal mammalian repertoire, contributing to adaptation and survival. However, when aggression is excessive and does not correspond to the context and the stimulus, it is considered pathological [1,2,3,4,5]. The pathological form of aggression is primarily aimed at obtaining pleasure and emotional reward from an attack [6,7]. It has been shown that access to expression of aggression can serve as a positive reinforcement, and aggression can be accompanied by addiction similar to drug addiction [8].

Aggressive behavior is accompanied by dopamine release in the brain’s reward circuitry, such as the nucleus accumbens (NAc) and ventral tegmental area (VTA) [9,10]. Furthermore, long-term aggressive experience leads to persistent activation of the dopaminergic system [11], including changes in the sensitivity of D1 and D2 dopamine receptors [12,13]. Animal studies have shown that previous fighting experience facilitates aggressive behavior in subsequent encounters and leads to the development of heightened aggression in most species [14,15]. This process can also result in pathological patterns of aggression, such as attacking females, pups, or immobilized males; persisting with attacks despite clear submissive signals from the opponent; and inflicting severe injuries [11,13,16,17,18,19,20]. It has been shown that even after the aggressive encounters have ended, mice maintain a high level of aggression for a long period of time [14,20,21].

The relationship between aggression and the immune system is complex. On one hand, acute inflammation suppresses dopamine release in the nucleus accumbens [22] and reduces aggression in animals [23,24,25,26,27]. On the other hand, it has been shown that the immune system is more active in aggressive animals, as evidenced by their slowed tumor development [28,29] and metastasis formation [29]. Aggressive mice also demonstrate a stronger immune response to acute pro-inflammatory stimuli [30,31,32].

Microglia, the brain’s resident immune cells, are a key mediator between the immune and central nervous systems [33,34]. Beyond responding to inflammation, they also regulate neural plasticity and social behavior [33,34,35,36]. Furthermore, recent research has shown that microglia are activated in response to aggressive behavior [37,38].

Most research on the relationship between inflammation and aggressive behavior is limited to studying acute effects. Chronic inflammation could lead to impairments in its resolution and have long-term consequences. The delayed effects of inflammation and their influence on aggressive behavior remain poorly investigated. In this study, we assessed the influence of the delayed effects of chronic LPS-induced inflammation on the behavior and neuroimmune parameters of mice with prolonged experience of aggression, developed in a sensory contact model [39]. We examined how chronic inflammation affects the level and pattern of aggression, specifically the manifestation of its pathological form (using a test with an immobilized conspecific). To evaluate immune status, we analyzed changes in microglial profile, blood cell subpopulations, and the expression of inflammatory markers in two brain structures associated with aggressive behavior: the hypothalamus and nucleus accumbens. Additionally, we analyzed the expression of dopaminergic system genes in these regions to identify potential markers associated with the development of pathological aggression.

## 2. Results

### 2.1. Prolonged Experience of Aggression Leads to the Development of Three Distinct Patterns of Aggressive Behavior

Positive fighting experience was formed after 30 days of daily aggressive confrontations with a weaker C57Bl/6 male mouse, leading to behavioral variations in the display of aggressive behavior in male CD1 mice. During the experiment, we analyzed the mice’s aggressive behavior using two tests: “Aggressive Confrontation with a Moving Partner” (which measured hostility toward a freely moving C57Bl/6 partner) and “Pathological Aggression” (which measured aggression toward an immobilized conspecific). Figure 1A illustrates the design of the experiment. The behavioral analysis was conducted at three time points: P1—the beginning of aggressive confrontations, P2—30 days of aggressive experience, and P3—after the end of the aggressive confrontation and repeated LPS treatment.

We used k-means cluster analysis for the main parameters in two aggressive tests (“latency time before the first attack” and “attack duration”) and divided mice into three subgroups.

“Low Aggression mice” (LA, 24% of mice, n = 11): mice that reduced aggression to a moving partner during the experiment and did not exhibit pathological aggression.

“Non-pathological Aggressors” (NA, 40% of mice, n = 18): mice that exhibited strong aggression towards their partner during the experiment and did not exhibit pathological aggression.

“Pathological Aggressors” (PA, 36% of mice, n = 17): mice that exhibited strong aggression towards their partner during the experiment, and pathological aggression towards an immobilized partner increased with aggression experience.

Mice in the identified subgroups demonstrated significant differences in aggressive behavior in both tests and also changed behavior differently over time (for factorial analysis results, see Appendix A).

In the “Aggressive confrontation with a moving partner” test, mice from the “Low Aggression” (LA) group showed a significant decrease in their aggressive responses over time. At the first time point (P1), LA mice already displayed lower aggression, showing the longest attack latency (*p* < 0.05 vs. PA). After 30 days of experience (P2), this phenotype became more pronounced: the LA group demonstrated the shortest attack time (*p* < 0.001 vs. NA and PA) and the longest attack latency (*p* < 0.01 vs. NA and PA) of all groups (Figure 1C). Furthermore, within the LA group, aggression decreased significantly over time, with a reduction in attack time and an increase in attack latency from P1 to P2 (*p* < 0.001 for both; Figure 1C). Consistently, in the “Pathological aggression” test, the LA group maintained the lowest level of aggression across all testing points (P1–P3) (Figure 1B).

The “Non-pathological Aggressors” (NA) group displayed high aggression in the “Aggressive confrontation with a moving partner” test and did not change it over time (P1 vs. P2). In the “Pathological aggression” test, similarly to the LA mice, they demonstrated a low level of aggression to an immobilized partner throughout the duration of the experiment (Figure 1B).

The “Pathological Aggressors” (PA) group demonstrated consistently high and escalating aggression. In the “Aggressive confrontation with a moving partner” test, PA mice exhibited the shortest attack latency at P1 (*p* < 0.01 vs. LA) and after 30 days of experience at P2 (*p* < 0.05 vs. LA), with no significant decrease in their aggressive response (Figure 1C). A similar hyper-aggressive phenotype was observed in the “Pathological aggression” test. Here, the PA group displayed the highest attack time and shortest latency from P1 (*p* < 0.05–0.001 vs. all other groups; Figure 1B) and maintained this status at P2 and P3 (*p* < 0.001 vs. NA/LA for both), also showing heightened locomotor activity (*p* = 0.01 vs. LA at P2 and P3). Furthermore, their aggression to immobilized partners intensified over time, as evidenced by a significant increase in attack time (*p* < 0.01 P1 vs. P2, *p* < 0.001 P1 vs. P3) and a decrease in attack latency (*p* < 0.05 P1 vs. P2/P3) across time points (Figure 1B).

Analysis of aggressors as a whole (combining all subgroups) showed that they displayed a significant increase in locomotor activity in the “Pathological aggression” test between time points, unlike control mice, which did not change their locomotor activity. In the combined group of aggressors, locomotor activity increased between points P1–P2 (*p* = 0.0012) and P1–P3 (*p* < 0.001, Figure 1B). Also, NA and PA saline-treated groups, as well as combined group of aggressors, demonstrated significantly higher locomotor activity compared to a saline-treated control at P3 (NA *p* = 0.008, PA *p* < 0.001, combined group’ *p* < 0.001; Figure 1B).

Aggressive motivation was assessed using the “Partition test” in the home cage with a familiar partner immediately before the confrontation at P2. It is measured in the time aggressors spend close to the partition [40]. Significant differences were observed between subgroups, with the PA group spending significantly more time near the partition than the LA group (*p* < 0.05; Figure 1D). This elevated motivation in PA mice was accompanied by increased locomotor activity in the “Pathological Aggression” test.

### 2.2. Mice Demonstrate Increased Impulsivity, Impaired Decision-Making, and Risk Assessment After Prolonged Experience of Aggression

To assess the effect of aggressive experience on anxiety levels, we evaluated the behavior of mice in the elevated plus maze test (Figure 1E). Aggressive experience significantly reduced the total time spent in the open arms compared to the control group (*p* < 0.001 Control vs. Aggressors; *p* = 0.014 Control vs. LA; *p* = 0.016 Control vs. NA; *p* < 0.001 Control vs. PA), with the most pronounced effect for the PA group, and increased the total time spent in the closed arms compared to the control group (*p* = 0.03 Control vs. Aggressors). In addition, only the “Pathological Aggressors” group spent significantly more time in the center of the maze compared to the control group (*p* < 0.001 Control vs. PA). Aggression experience also led to increased number of poking from the closed arms (*p* < 0.01 Control vs. Aggressors; *p* < 0.05 Control vs. LA; *p* < 0.05 Control vs. PA). Increased time spent in the center of the maze often correlates with increased impulsivity [41] and decision-making impairment [42]. Thus, the “Pathological Aggressors” group, in addition to social behavior impairments, also demonstrate increased impulsivity and impaired decision-making.

### 2.3. Chronic LPS Treatment Fails to Reduce Aggression and Has No Effect on Anxious Behavior

Two days after the last LPS injection, the light-dark box test was performed to assess the effects of inflammation on anxiety behavior (Figure 1F). In control mice, LPS treatment induced a trend towards increased latency time (*p* = 0.07 Control-Saline vs. Control-LPS) and a significant decrease in the number of light box entrances (*p* = 0.05 Control-Saline vs. Control-LPS) Aggressive experience led to a significant decrease in the number of poking from the dark box, an effect observed in all aggressor subgroups compared to controls (*p* < 0.001 Control vs. Aggressors; *p* < 0.001 Control vs. LA; *p* < 0.001 Control vs. NA; *p* < 0.001 Control vs. PA).

Thus, two days after the last LPS treatment, control animals exhibit increased latency and a reduced number of entrances to the light compartment, which could indicate a decrease in locomotion. We hypothesize that parameters may be indirect markers of sickness behavior. Aggressive mice reduced the number of peeks from the dark box, indicating a decreased risk assessment [43], and did not exhibit indirect signs of sickness behavior, suggesting that the inflammatory response had already ended in the aggressors by this time.

The effects of LPS treatment on aggression were assessed in the “Pathological aggression” test 4 days after inflammation (P3). Chronic LPS treatment fails to reduce aggression in all subgroups. Moreover, NA and PA groups, which received LPS, demonstrated significantly higher locomotor activity compared to a control group that received LPS (*p* = 0.04 and *p* = 0.002, respectively; Figure 1B).

### 2.4. Aggressive Mice Show Faster Resolution of Inflammation Five Days After Chronic LPS Treatment

The mice showed signs of illness during the first three to four days of LPS treatment, including decreased reaction to our manipulations, immobility, hunching posture, body weight loss, swelling, and diarrhea. By the fifth day, the mice’s health had returned to normal, and they started to gain weight. Nevertheless, the mice had recovered only 95% of their starting weight at the end of the experiment five days after the last LPS (Figure 2A).

The relative spleen weight was influenced by factors: “LPS treatment” [F(2,48) = 28.6, *p* < 0.001], “aggression” [F(2,48) = 4.77, *p* < 0.033], and the interaction between “LPS treatment” and “aggression” [F(2,48) = 5.21, *p* < 0.026], but post hoc analysis did not find any significant differences between aggressors and control with different treatment. LPS treatment resulted in spleen enlargement in aggressors (*p* < 0.001) but not in controls (Figure 2B).

The “aggression” factor was significantly affecting the relative weight of the adrenal glands compared to the control [F(1,56) = 20.8, *p* < 0.001], but pair-wise comparisons were not significant. The “aggression” factor also affected adrenal hyperplasia induced by the LPS [F(1,56) = 5.6, *p* < 0.05]: the weight of the adrenal glands was significantly increased in the control group with LPS treatment compared to the aggressors (Figure 2B).

Thus, the experience of aggression resulted in an enlarged spleen after LPS administration, decreased adrenal weight, and inhibited LPS-induced adrenal hyperplasia. Subgroups of aggressors did not differ in these parameters and responded equally to LPS treatment (Appendix A).

To assess the effects of long-term aggressive experience on the peripheral immune system, we performed a complete blood count at multiple time points. Repeated measures ANOVA revealed that aggressive mice exhibited a significant increase in white blood cell (WBC) count (*p* < 0.001 P1 vs. P2), as well as lymphocyte (*p* < 0.001 P1 vs. P2) and monocyte (*p* < 0.01 P1 vs. P2) counts after 30 days of aggressive encounters (Figure 2C). After LPS-induced inflammation (P3), factor “aggression” affected WBC [F(2,48) = 20.7, *p* < 0.001] and lymphocyte percentage [F(2,48) = 4.29, *p* = 0.044], although for lymphocyte percentage no significant differences between aggressors and control with different treatment were found (according to Tukey HSD post hoc test). LPS injections led to an increase in WBC in control (*p* < 0.001 P2 vs. P3), but not in aggressors, who do not demonstrate any inflammatory reactions in blood at this time point (Figure 2D), regardless of the subgroup of aggressors (Appendix A). Thus, prolonged experience of aggression led to an increase in leukocytes, in particular lymphocytes, in the blood of mice.

In order to link observed changes in the peripheral immune system to the state of brain immunity, we assessed the changes in the brain immune cells—microglia. In this study we analyzed the proportion of resting microglia (P2RY12+/CD11b+/CD45int) and active microglia (P2RY12+/CD11b+/CD45high; Figure 2E). Both aggressive experience [P2RY12+/CD45int/CD11b+ F(2,53) = 33.3, *p* < 0.001; P2RY12+/CD45hi/CD11b+ F(2,53) = 6.95, *p* < 0.01] and LPS treatment [P2RY12+/CD45hi/CD11b+ F(2,53) = 25.61, *p* < 0.001; P2RY12+/CD45int/CD11b+; F(2,53) = 27.3, *p* < 0.001] affected these populations. Aggressive experience did not affect microglial populations in the basal state, but it did affect their reactivity to LPS treatment. Aggressors had a higher percentage of resting microglia (*p* < 0.01 Control LPS vs. Aggressors LPS) and a lower percentage of active microglia (*p* < 0.001 Control LPS vs. Aggressors LPS) compared to the control group. Five days after LPS administration, microglia in control mice remained activated (*p* < 0.001 Saline vs. LPS), whereas in aggressors, the number of active microglia had already decreased (tendency, *p* = 0.076 Saline vs. LPS) (Figure 2F). We also did not find differences between subgroups of aggressors (Appendix A). Consistent with results in the peripheral immune system, in the brain aggressors demonstrate faster resolution of LPS-induced inflammation.

### 2.5. Expression of Inflammatory Genes After LPS Treatment Is More Pronounced in the Hypothalamus than in the NAc

We evaluated the level of gene expression involved in the inflammatory response in the hypothalamus and the nucleus accumbens, as both structures are parts of aggression circuitry: the ventromedial hypothalamus is a key structure for the execution of aggressive behavior, and the NAc is a key structure in the rewarding aspects of aggression [8]. Moreover, peripheral LPS-induced inflammation reaches the brain via hypothalamic structures [44]. Since gene expression did not differ between subgroups of mice (see Appendix A), we analyzed changes in the aggressor group as a whole (for factorial analysis results, see Appendix A).

We found that the effect of inflammation in the aggressive mice was more pronounced in the hypothalamus than in the NAc, although the direction of the changes was similar (Figure 3).

In the hypothalamus, LPS non-specifically increased the expression level of the pro-inflammatory genes *Aif1* and *Il1b* (AIF1 is an activated microglia marker, and cytokine IL-1β is the first step in the NF-κB inflammatory pathway) in both control and aggressor groups (*p* < 0.001 for both), whereas in the NAc a significant LPS effect was observed only for the *Il1b* gene in the control group (*p* = 0.001 Control-LPS vs. Control-Saline).

Notably, in mice with aggressive experience, the expression of the *Gfap* gene, which is a marker of activated astrocytes, and the *Trem2* gene, the product of which suppresses microglial hyperactivation during inflammation and provides neuroprotection, was increased in both structures (*p* < 0.03 for all), and in the NAc of saline-treatment mice, *Trem2* expression was significantly higher in aggressors (*p* = 0.01 Agg-Saline vs. Control-Saline). The expression level of the *Cxcl10* gene (CXCL10 is a pro-inflammatory cytokine) was increased only in the LPS-treated control in both structures (*p* < 0.005 for hypothalamus, *p* < 0.05 for NAc) but did not change between the differently treated aggressor groups. Thus, at the gene expression level we also see that the aggressive experience modulates the response to the inflammation, leading to a faster resolution. However, the expression level of *Traf6* (TRAF6 in microglia activates the NF-κB transcription factor) was higher in LPS-treated aggressors (*p* = 0.05 Aggressors-LPS vs. Control-LPS), compared to control.

Additionally, correlation analysis also revealed a link between *Il1b* expression and aggression: in the aggressive mice, *Il1b* expression in the NAc negatively correlates with indicators of pathological aggression at the P3 time point (for *Il1b* expression vs. attack time, R = −0.38, *p* = 0.02; Figure 4A).

### 2.6. Expression of Genes Related to the HPA and Dopaminergic System Is Modulated Differently in Hypothalamus and Nucleus Accumbens and Is Affected by Aggressive Experience, but Not LPS Treatment

We also assessed changes in the expression of genes related to the HPA and dopaminergic systems in the hypothalamus and nucleus accumbens, since those systems are notably involved in the aggressive behavior [11,45,46] and are also known to modulate the inflammatory response [47,48,49].

We found that the glucocorticoid receptor *Nr3c1* gene expression level in the NAc was significantly lower in the aggressive mice compared to the control (*p* < 0.001). Additionally, changes in the expression of the D1 dopamine receptor gene *Drd1* were oppositely directed in the studied structures: the expression level was increased in the hypothalamus but decreased in the NAc in aggressive mice compared to the control (*p* = 0.03 for both). Surprisingly, the expression of other genes related to the HPA axis (*Crh*, *Crhr1*, *Fkbp5*) and the dopaminergic system (*Drd2*, *Th*, *Maoa*, *Ppp1r1b*, *Slc6a3*) was not dependent on the aggressive experience or LPS treatment in our experiment. The identified subgroups of aggressors (LA, NA, PA) did not differ in the expression levels of the genes we analyzed, and the direction of changes compared to the control group was the same for all subgroups (Appendix A).

Correlation analysis showed a strong positive correlation between the expression of *Drd1* and *Drd2* genes in NAc in aggressors (R = 0.83, *p* < 0.001), but not in controls (R = −0.09) (Figure 4B).

## 3. Discussion

Our study demonstrates that prolonged aggressive experience in male CD1 mice does not merely alter the animals’ behavioral profile but forms a unique neuroimmune profile that modulates the brain’s response to systemic inflammation. Aggressive mice show an LPS-induced inflammatory response characterized by faster resolution of microglial activation and altered expression of key immune and dopaminergic system genes. This indicates a long-term adaptation of neuroimmune mechanisms underlying the modified social behavior.

A prolonged experience of aggression causes a fundamental reorganization of behavioral patterns in animals. Previous work with this model has established that repeated aggression leads to significant alterations in mouse behavior, including elevated anxiety-like behavior [50], impaired social recognition [13], and hyperactivity with enhanced stereotypic behaviors [51]. In the present study, we further demonstrate that aggressive mice display increased motor impulsivity, impaired decision-making, and reduced risk assessment.

Kudryavtseva (2020) [11] proposed that in some mice chronic aggression leads to behavioral pathology characterized by a reduced attack threshold and attacks on pups and females, thereby disrupting the adaptive function of aggression. In male C57Bl/6 mice “learned aggression” is observed, i.e., direct attacks are replaced with aggressive grooming and indirect hostile behavior. In our study we used a different mouse strain, CD1, which is known to exhibit high levels of aggression toward different categories of social targets [52] and which does not typically display aggressive grooming [51,53]. We used a test for pathological aggression, adapted from Natarajan and Caramaschi [20,54], based on the aggressor’s interaction with a neutral social stimulus (an immobilized male). We demonstrated that 36% of aggressors of this strain developed pathological aggression: they attacked an anesthetized male, which posed no threat. The remaining males exhibited aggression only in response to provocative behaviors from the partner. While 40% of males displayed intense aggression regardless of the partner’s active or submissive behavior (“Non-pathological Aggressors”), 24% reduced their aggressive behavior when the partner demonstrated submissive behavior (“Low Aggressive mice”).

The “Pathological Aggressors” group exhibits high levels of aggression toward both the immobilized male and the active-moving partner. This high aggression toward the active-moving partner is present from the very first confrontations and does not diminish over time, even when the partner begins displaying submissive behavior. This group of aggressors showed the highest level of aggressive motivation, which we assessed before confrontation by their reaction to a partner behind a partition in a common cage, and also demonstrated elevated locomotor activity during the “Pathological aggression” test, reflecting extreme agitation upon contact with the immobilized partner. Finally, the strong aggressive behavior in “Pathological Aggressors” is accompanied by high impulsivity, reduced risk assessment, and impaired decision-making. Thus, the “Pathological Aggressors” group is characterized by a distinct pathological profile: their aggression is uncontrolled and maladaptive, indicating dysregulation in the control of aggressive behavior.

“Non-pathological Aggressors” display high aggression toward a moving partner and elevated agitation in the “Pathological aggression” test after the 10-day deprivation of aggression. However, in contrast to the PA group, they do not exhibit aggression towards an immobilized mouse and also do not demonstrate any decision-making impairment.

Unlike other groups, “Low Aggressive mice” show a progressive decline in aggression toward a moving partner and exhibit no pathological aggression or high agitation. It is important to note that mice from the LA group nevertheless exhibited aggressive behavior. In all confrontations, they displayed dominant actions (attacks or aggressive grooming) and attacked partners if provoked. Overall, during the 30-day period of confrontations, the LA mice attacked their partners in at least half of the cases. Moreover, they share the decision-making impairment seen in the PA group.

These behavioral patterns are stable characteristics of the CD1 mouse strain, as we have observed consistent results in our previous work with this experimental model [21].

It is well established that peripheral inflammation can affect an individual’s psychological state and is a risk factor for the development and exacerbation of mental illness; for example, it can induce depressive symptoms, increase irritability and anxiety, and cause working memory impairment [55,56,57]. Inflammation is also known to suppress dopamine release in brain regions responsible for motivation and pleasure, such as the nucleus accumbens [22]. Therefore, we hypothesized that aggressors with different patterns of aggressive behavior would respond differently to induced inflammation, and specifically mice with pathological aggressive behavior would be more sensitive to it.

In this study we were unable to detect significant differences between the aggressor subgroups in terms of immune response parameters and gene expression in the hypothalamus and nucleus accumbens (Appendix A). The similar changes in the parameters we assessed across all aggressors may suggest that the prolonged manifestation of aggression, regardless of its intensity level, exerts a significant influence on the immune response. A more complete understanding of the molecular basis of pathological aggression will likely require examining its interplay with other critical neural systems, notably the opioid and serotonin pathways.

Therefore, in what follows we will discuss the group of aggressors as a whole, without dividing it into subgroups.

LPS treatment induced robust sickness behavior in both aggressor and control mice, characterized by reduced activity and responsiveness (including immobility and diminished reaction to handling), diffuse piloerection, a hunched posture, swelling, and diarrhea [58]. The sickness state was accompanied by significant weight loss in the first few days during LPS injection. Five days after LPS was discontinued, the mice body weight had nearly returned to normal, and all sickness symptoms had resolved. Also, LPS treatment ultimately had no significant effect on anxiety-like behavior, levels of pathological aggression, or locomotor activity in either the controls or aggressors. Furthermore, this effect was consistent across all previously identified subgroups of aggressors.

Previous research has shown that a single LPS injection can reduce aggression in dominant and aggressive individuals [23,25,26]. However, those studies assessed behavior immediately or within hours after injection, during the acute inflammatory phase. In our study, behavioral testing was conducted after the sickness behavior had fully subsided. This is consistent with other studies of repeated LPS treatment, which also found no delayed effects of chronic LPS treatment on behavior (up to 50 days later) [59]. Therefore, we conclude that the LPS-induced reduction in aggression is a transient and reversible phenomenon, tightly linked to the acute sickness state, and does not produce long-term changes in the behavioral phenotype.

Previous studies that focus on the characteristics of the immune response in aggressors have shown that the experience of aggression leads to a stronger activation of T-cell immunity: aggressors showed a significantly higher number of active T-cells in the spleen and bone marrow compared to controls on the 5th day after immunization [60,61,62]. This is supported by our data on higher spleen weight after LPS exposure only in aggressive mice, indicating lymphocyte activation and proliferation. Furthermore, the blood cell profile in the aggressors is already indicative of the resolution phase of inflammation. This is evidenced by the absence of leukocytosis in the LPS-treated aggressors, unlike in the control group, which maintained an elevated leukocyte count.

A different dynamic of the LPS response between aggressive and non-aggressive mice was previously demonstrated by Idova (2016) [31] in the production of the pro-inflammatory cytokine TNF by splenocytes. Aggressive individuals exhibit a stronger and more rapid immune response to acute LPS treatment, marked by intense early TNF production that declines faster than in controls [30], suggesting an earlier onset of the inflammation resolution phase. This heightened acute phase immune reactivity (6–24 h) in aggressors is further supported by splenic hyperplasia [26], enhanced NK- and T-cell responses [32], and increased susceptibility to autoimmune conditions in aggressive wild rats [63]. Thus, aggressive behavior induces a distinct immunological phenotype characterized by robust initial immune activation followed by accelerated resolution of inflammation. This pattern is evidenced by our findings of sustained splenic activity alongside normalized peripheral blood parameters by day 5 post-LPS administration.

This stronger response to acute inflammation can be explained by a primed hyperinflammatory phenotype in aggressors. It is known that chronic stress primes innate immune responses in mice [64]. While this specific process has not been demonstrated for aggressive behavior, prolonged aggressive experience also constitutes a stressor for the organism [65], allowing us to hypothesize similar underlying mechanisms. In our work, we showed that 30 days of aggression experience increases the number of leukocytes (lymphocytes and monocytes) in peripheral blood. Although this increase cannot be classified as the development of an inflammatory process since the values remain within the normal range for mice, it nevertheless indirectly indicates a state of low-grade inflammation. Other studies also found that aggressors have elevated spontaneous levels of the pro-inflammatory cytokines IL-2 and IFN-γ [30], indicating a heightened baseline immune activation. This may explain their reduced susceptibility to tumors: aggressive mice develop smaller tumors [66] and exhibit higher NK cell cytotoxicity [28], as well as fewer metastases [29]. Thus, prolonged aggressive experience is associated with low-grade systemic inflammation and an enhanced defensive capacity of the immune system.

Microglia are the primary cells of the immune system in the brain, but their relationship with the aggressive phenotype is poorly understood [37]. In aggressive mice, microglia are activated and proliferate in the anterior cingulate cortex [38] and midcingulate cortex [67]. In the dentate gyrus, microglia take a hyper-ramified form while simultaneously decreasing in number, possibly due to apoptosis resulting from microglia hyperactivation [68]. These data suggest that microglia activation and subsequent neuroinflammation in specific brain regions may be associated with increased aggression in male mice.

Our study shows that prolonged aggressive experience does not alter baseline microglial populations but significantly modifies their inflammatory response. Mirroring peripheral results, control mice, but not aggressive mice, showed an increased proportion of activated microglia five days post-LPS. This indicates that microglia in aggressive animals had already returned to a resting state, demonstrating that aggression experience accelerates both the immune response and its resolution in the brain.

Gene expression analysis further supported this assumption. In the nucleus accumbens (NAc), expression of the *Aif1* gene, whose protein product Iba1 serves as a microglial activation marker, was elevated in the LPS-treated group only in control, but not in aggressors. A similar pattern was observed for expression of the chemokine *Cxcl10*, which is produced by activated astrocytes and functions as a microglial activator [69]. CXCL10 is classified among “late-phase cytokines” and typically remains elevated for an extended period following systemic LPS challenge [70]. However, in aggressors, its expression had already returned to baseline levels.

In the hypothalamus, LPS increased expression of the pro-inflammatory genes *Aif1* and *Il1b* similarly in both control and aggressive mice. This regional contrast underscores the brain’s differential vulnerability to inflammation. The hypothalamus, which contains regions lacking a blood–brain barrier, acts as a primary gateway for peripheral inflammatory signals [44], explaining the persistent inflammation observed there five days post-LPS. Conversely, in the nucleus accumbens—a key motivation and reward center—these inflammatory markers had resolved in aggressors by the same time point.

Typically, microglial activation is rapidly suppressed to prevent secondary neuronal damage [71]. The microglial receptor TREM2 facilitates this negative feedback, promoting a switch to an anti-inflammatory phenotype upon binding apoptotic cell lipids or bacterial ligands [72,73]. We found that aggressive mice exhibited twofold higher *Trem2* expression in the NAc compared to controls, with a similar increase in the hypothalamus. We hypothesize that heightened gene expression leads to enhanced TREM2 level and this, in turn, may contribute to the accelerated inflammatory resolution observed in these mice.

Our study found no strong interaction between the development of pathological aggression and the overall neuroimmunological profile. Neither microglial populations nor the expression of most inflammatory response genes correlated with different aggression patterns. The exception was *Il1b* gene expression in the nucleus accumbens, which positively correlated with pathological aggression levels—an association absent in the hypothalamus (Figure 4A). The relationship between IL-1β and aggression has been previously established [74], with studies showing that high aggression levels accompany lower IL-1 β concentrations in the dorsal raphe nuclei, but not in other brain regions. Thus, we hypothesize that specifically low levels of IL-1 β in key structures of the reward system may weaken inhibitory control over aggressive impulses, thereby facilitating the development of pathological behavior.

The inflammatory response is strongly modulated by the HPA axis, though findings on glucocorticoid activation during aggression remain contradictory. One hypothesis proposes a U-shaped relationship between aggression and HPA axis reactivity [45], supported by studies of rodent strains with contrasting HPA reactivity but comparable high aggression [75,76,77].

In our previous work [53], we found that chronic experience of aggression reduces relative adrenal weight. The current study shows this reduction persists for 12 days after ending aggressive confrontations, suggesting potential adrenal exhaustion. Accordingly, aggressors showed a blunted adrenal response to LPS compared to controls. We also showed that corticosterone levels and relative adrenal mass in aggressors normalized only 30 days after the end of confrontations [21].

In the hypothalamus, prolonged aggression did not alter glucocorticoid receptor (GR) gene (*Nr3c1*) expression either immediately after the aggressive experience [53] or at 12 days (present study) or 30 days following the end of aggressive confrontations [21]. This indicates preserved hypothalamic sensitivity to negative feedback in aggressors. Our interpretation is further supported by the absence of changes in *Fkbp5* expression, a gene encoding the FKBP51 co-chaperone that regulates GR cytoplasmic retention and inactivation [78]. Aggression experience did, however, reduce expression of CRH secretion inhibitors (*Crhbp*, *Crhr1*), which would be expected to increase CRH signaling tonus [79] in hypothalamus. Following the end of aggressive confrontations, *Crhr1* expression normalized (present study), and *Crh* expression levels decreased by day 30 [21]. In contrast, in the nucleus accumbens, we observed reduced *Nr3c1* expression after the cessation of aggressive interactions, with gradual recovery by day 30 [21]. This suggests a transient decrease in glucocorticoid sensitivity within the reward system during the post-aggression period. Consistent with our findings, other studies report no GR expression changes in the hypothalamus of highly aggressive rats [45], hippocampus of aggression-selected rats [76], or GR protein levels across brain regions [80].

Under our experimental conditions, chronic LPS treatment did not affect HPA axis gene expression in either controls or aggressors. This lack of effect is likely due to the extended post-LPS interval in our study, which allowed sufficient time for the recovery of basal HPA axis gene expression. This interpretation is consistent with studies in rats showing that reduced *Crh* mRNA in the hypothalamic paraventricular nucleus observed 24 h after a final LPS injection [81] normalizes over time.

Acute and repeated aggression are associated with sustained activation of the dopaminergic system and increased dopamine levels in the hypothalamus and limbic system [9,46,82]. In male C57Bl/6 mice with prolonged aggressive experience, sensitivity to dopamine receptor antagonists (haloperidol and SCH-23,390) is reduced [12,13], and expression of D1 and D2 dopamine receptor genes in the NAc decreases, followed by a subsequent increase after the end of aggressive confrontations [11,83,84]. In our work with CD1 mice, a similar but temporally delayed dynamic was observed for the D1 receptor gene: expression decreased on day 12 (present study) and increased by day 30 of deprivation [21], while D2 receptor gene expression did not change. The differences in dynamics may be related to the mouse strain and their level of manifested aggression, as well as the method of NAc isolation, since we used a more precise method of targeted zone extraction from frozen sections with histological control of brain structures. Importantly, the observed D1 receptor dynamics were bidirectional: in the hypothalamus, its expression demonstrated the opposite pattern to the NAc (increasing on day 12 (this study) and decreasing by day 30 of deprivation [21]). Changes in hypothalamic D1 receptors after prolonged aggressive experience likely reflect an adaptive reorganization to normalize CRH system activity (by inhibiting POMC neurons in the PVH that synthesize CRH) [85] and could even possibly be involved in the modulation of the immune response [47,48]. Dopamine, acting through D1 [86] and D2 [87,88,89] receptors, directly suppresses inflammation by inhibiting key pro-inflammatory pathways, which could explain the faster resolution of the inflammatory response to LPS in aggressive mice, but this notion requires additional research.

Both acute and chronic inflammation reduce dopamine levels [90,91] and its binding to D2 receptors in the striatum, while suppressing D2 receptor activity on astrocytes and thereby enhancing neuroinflammation [89,92]. This process leads to inflammatory anhedonia, as cytokines disrupt dopamine synthesis, transport, and reuptake [22]. The suppression of the dopaminergic system also accounts for the observed reduction in aggression following acute LPS administration [23,25]. However, the effects of LPS are transient. In our experimental paradigm, five days post-LPS administration, we detected no effect of inflammation on the dopaminergic system in the hypothalamus: neither on the expression of dopaminergic receptor genes nor on genes encoding dopamine metabolic enzymes (*Maoa* and *Th*).

Importantly, our findings, consistent with previous research [21], demonstrated a strong positive correlation between *Drd1* and *Drd2* gene expression specifically in the NAc of aggressors, but not in controls. This indicates coordinated regulation of these receptors within this brain structure and provides transcriptional-level confirmation of their previously documented colocalization and heterodimer formation [93,94]. Since activation of D1-D2 heteromeric complexes plays a crucial role in long-term neuroplasticity [95], the tight coupling in their gene expression may represent the molecular foundation for the persistent reward system alterations characteristic of pathological aggression [21].

Thus, this study demonstrates that prolonged aggressive experience induces persistent alterations across behavioral, neuroimmune, and neuroendocrine systems in male CD1 mice. Aggressors develop a distinct neuroimmune profile characterized by accelerated resolution of both systemic and central inflammation following LPS challenge. This earlier onset of the inflammation resolution phase manifests through quicker normalization of microglial activation states, peripheral blood parameters, and expression of key pro-inflammatory genes in the nucleus accumbens. The observed adaptive immune response is possibly modulated by an increased *Trem2* expression and dopaminergic signaling, but this preposition requires additional research. Notably, while a subset of mice developed maladaptive pathological aggression, the neuroimmune profile showed minimal correlation with behavioral subgroups except for NAc *Il1b* expression levels. Similar immune changes in all subgroups of aggressors indicate that prolonged aggression itself, regardless of degree, significantly impacts immunity. Collectively, these findings reveal complex mechanisms of long-term plasticity underlying the relationship between aggressive experience, brain immune function, and dopaminergic system activity.

It is also important to acknowledge several methodological constraints that shape the interpretation of our results and suggest priorities for subsequent research. Our findings are constrained by the measurement of inflammatory markers at the gene expression level, the lack of temporal tracking of the immune response and the absence of specific tests for sickness behavior. To build upon this work, subsequent studies should incorporate more time points, quantify cytokine and chemokine levels at the protein level in both the periphery and brain, and perform immunohistochemical analyses of microglial and astroglial morphology. These approaches would greatly increase the translational relevance of the research and provide a more comprehensive elucidation of immune system dynamics in aggressive animals. A more holistic model would also require examining the interplay with other critical neural systems implicated in aggression, notably the opioid and serotonin pathways.

## 4. Materials and Methods

### 4.1. Animals

The experiment was conducted on male CD1 mice aged 2.5–3 months. The mice were maintained under standard conventional vivarium conditions at the Institute of Cytology and Genetics, Siberian Branch of the Russian Academy of Sciences (12:12 h light/dark cycle with lights off at 19:00, food and water available ad libitum). Male C57Bl/6 mice aged 2.5–3 months served as cage partners. The animal study protocol was approved by the Ethics Committee of the Institute of Cytology and Genetics, SB RAS (protocol №175/1, 24 July 2024) in conformity with European Communities Council Directive 210/63/EU of 22 September 2010.

### 4.2. Prolonged Aggressive Experience Model

Aggressive behavior was developed using the sensory contact model [39] with modifications [22,53]. Pairs of male mice (CD1 mouse-C57Bl/6 mouse) were housed in cages (28 × 14 × 10 cm^3^) divided by a transparent perforated partition, allowing visual, auditory, and olfactory interaction while preventing physical contact.

Testing started after two days of acclimation to the housing conditions and sensory contact. The partition was removed daily between 14:00 and 17:00 for 10 min to enable aggressive interactions. Dominance of CD1 mice was established within the first three days. CD1 mice consistently attacked, bit, and pursued C57Bl/6 mice, which in turn displayed defensive behaviors (sideways postures, upright positions, evasion, supine postures, or freezing). If intense aggression persisted beyond one minute, aggressive encounters were terminated by replacing the partition. To prevent habituation, C57Bl/6 mice were rotated daily between cages after each confrontation, being placed in unfamiliar cages with unfamiliar dominant CD1 males behind partitions until the next session. This ensured CD1 males remained in their original cages but encountered new partners daily, thereby stimulating aggressive behavior upon partition removal. This procedure was performed once a day for 30 days.

### 4.3. Experimental Design

The experimental design is presented in Figure 1A. After the mice (n = 26) acquired 30 days of aggressive experience (see Methods 4.2), confrontations between the mice were discontinued, followed by a 7-day period of daily intraperitoneal LPS injections. LPS (500 μg/kg, volume–10 μL/g, solvent–saline, Escherichia coli serotype O55:B5, Sigma, St. Louis, MO, USA) or equivalent volume of saline was injected once a day between 09:00 and 10:00 AM. Mice were randomly assigned to LPS/Saline groups. There were 5–9 mice per injection group within each subgroup of Aggressors (for subgroup identification, see Results 2.1). Control animals consisted of age-matched CD1 mice without aggression experience, housed individually in experimental cages with partitions but no partners for 14 days. Control mice (n = 12) received the same treatment (LPS/saline). Exclusion criteria included visually assessed sickness behavior (decreased reaction to our manipulations, prolonged periods of immobility, swelling) prior to LPS administration.

The time points for behavioral tests and blood collection differed between the groups, as detailed below and in Figure 1A. Behavioral tests and blood sampling were performed no more than once per day.


*For Aggressor groups:*


P1 (baseline-aggression onset)—point before and at the beginning of aggressive confrontations (days 0–5 of the experiment)-blood collection for CBC, “Aggressive confrontation with a moving partner” test, “Pathological Aggression” test;

P2 (post-aggression-pre-inflammation)—point after 30 days of aggressive confrontations, before LPS treatment (days 26–30 of confrontations)-blood collection for CBC, “Partition Test” with “Aggressive confrontation with a moving partner,” Elevated plus maze, “Pathological Aggression” test;

P3 (post-inflammation)—point after chronic inflammation (2–5 days after the final LPS injection)-the Light-dark box test, “Pathological Aggression” test, blood collection for CBC.

Euthanasia—Day 41 (the day after final sampling, 5 days post-LPS).


*For Controls:*


P2 (pre-inflammation)—point before LPS treatment (day 0–3) (equivalent in timeline to Aggressors’ P2)-blood collection for CBC, Elevated plus maze, “Pathological Aggression” test;

P3 (post-inflammation)—point after chronic inflammation (2–5 days after the final LPS injection)-the Light-dark box test, “Pathological Aggression” test, blood collection for CBC;

Euthanasia—Day 14 (the day after final sampling, 5 days post-LPS).

### 4.4. Tissue Collection

Mice were euthanized under Avertin anesthesia (500 μL of 3% solution, intraperitoneal) followed by transcardial perfusion with PBS to clear the cerebral vasculature. Body weight, spleen weight, and adrenal weight were recorded for all animals. During the dissection, the hypothalamus was collected. One brain hemisphere was embedded in cryopreservative medium (Tissue-Tek O.C.T. Compound, Sakura Finetek, St. Torrance, CA, USA) and stored at −70 °C for later Nucleus accumbens isolation. The other hemisphere (with cerebellum removed) was used for flow cytometric analysis of microglial populations.

### 4.5. Behavioral Tests

#### 4.5.1. Pathological Aggression Test

To assess the development of pathological aggression at three time points (P1, P2, and P3), we analyzed the response to an anesthetized CD1 male mouse [20,54]. The subordinate partner was removed and replaced with an immobilized (anesthetized) CD1 male. The partition was then opened, allowing the subject mouse to freely interact with the anesthetized male for 3 min.

The following behaviors were quantified: latency time, attacks (offensive strikes, bites, and pursuit), and locomotor activity. For each behavior, we measured frequency and total duration. If no aggressive behavior was observed, the latency was recorded as 180 s (the total test duration), and other parameters were scored as zero.

#### 4.5.2. Aggressive Confrontation with a Moving Partner

Aggressive behavior toward a moving cage partner was assessed at P1 and P2 in the experimental cage, where the animals had previously experienced agonistic encounters. A subordinate male C57BL/6 mouse served as the interaction partner. The partition separating the mice was removed, and the 3 min test session was recorded on video. The following behaviors were subsequently quantified in the aggressive males: attacks, aggressive grooming, and latency time.

#### 4.5.3. Partition Test

To assess aggressive motivation toward a familiar cage partner, we quantified behavioral parameters near the partition separating the animals [40]. The test was conducted in the experimental cage at P2 immediately prior to the “Aggressive confrontation with a partner” test. During the 5 min “Partition Test,” mouse behavior was recorded before the partition was opened for aggressive confrontations. We evaluated parameters related to both aggressive behavior (number of approaches to the partition, total time spent near the partition) and individual behavioral patterns (vertical rearing, grooming).

#### 4.5.4. Elevated Plus-Maze Test

To assess the development of anxiety-related behavior after prolonged experience of aggression, we conducted an elevated plus-maze test. The test was conducted before LPS treatment (P2, 28 days of aggressive experience) and consisted of two open (25 × 5 cm^2^) and two closed arms (25 × 5 × 15 cm^3^), with two arms of each type opposite to each other and extending from a central platform (5 × 5 cm^2^). The maze was placed in a dimly lit room, and the following behavioral parameters were recorded during 5 min: open arm entrances (four paws in the open arm), closed arm entrances (four paws in the closed arm), and central platform entrances; total entrances; time spent in the open arms, closed arms, and central platform; and the number of poking when staying in closed arms (extending the head and body from the closed arm and quickly pulling back).

#### 4.5.5. Light-Dark Box Test

To assess anxiety levels after LPS treatment, the light-dark box test was conducted at P3. The apparatus consisted of two same-sized arenas (20 × 20 × 25 cm^3^). The light arena is made of white translucent plastic and brightly illuminated from below. The dark arena is made of opaque black plastic and covered with a lid. The arenas were connected by an opaque partition with a 7 × 7 cm^2^ opening.

At the start of the test, mice were placed in the light compartment. During the 5 min session, the following parameters were recorded: latency time before entering the dark box, number of transitions into the light box (all four paws), total time spent in the light box, number of transitions into the dark box (all four paws), total time spent in the dark box, and number of head pokes from the dark box into the light box.

### 4.6. Complete Blood Count

Blood was collected from the retro-orbital sinus between 10:00 and 11:00 AM. A 20 μL blood sample was immediately diluted in diluent (V-28D, Mindray MiniPack, Mindray, Huntingdon, UK) for subsequent analysis on a BC-2800 Vet hematology analyzer (Mindray, UK). The complete blood count included absolute and relative counts of lymphocytes, granulocytes, monocytes, and platelets.

### 4.7. Isolation of Microglia and Flow Cytometry Analysis

To obtain a perfused brain homogenate free of blood cells, mice were anesthetized with Avertin (500 μL of 3% solution, intraperitoneal) followed by transcardial perfusion with PBS. Microglia were isolated using a modified protocol based on a previously published protocol with some modifications [96]. One brain hemisphere was homogenized in PBS using a glass Dounce homogenizer, and the resulting cell suspension was purified from myelin and debris via centrifugation on a 37%/70% Percoll (Cytiva, Marlborough, MA, USA) gradient (800× *g*, 30 min). Cells collected from the 37%/70% interface were washed with 0.02% BSA in PBS, yielding approximately 3 × 10^5^ cells per sample. Cell suspensions were stained with monoclonal antibodies for 30 min on ice: CD45-PE-Cy7 (1 μg/10^6^ cells, clone 30-F11, E-AB-F1136H, Elabscience, Wuhan, China), CD11b-PerCP (1 μg/10^6^ cells, clone M1/70, E-AB-F1081F, Elabscience, Wuhan, China), and P2RY12-PE (0.2 μg/10^6^ cells, clone S16007D, 848003, Biolegend, San Diego, CA, USA). After washing, cells were analyzed on a BD FACSCanto™ II flow cytometer (BD Biosciences, San Jose, CA, USA), with a minimum of 100,000 events recorded per sample.

Gating strategy: the first two gates remove dead and doublet cells, based on their FSC/SSC plots. From live single cells, the next gate is cells positive for P2RY12+. The next gates were determined by the CD45+/CD11b+ positivity and differentiated by high and low expression of CD45+: resting microglia P2RY12+/CD45intermediate/CD11b+ and active microglia P2RY12+/CD45high/CD11b+ (Figure 2E).

### 4.8. Isolation of the Nucleus Accumbens

The nucleus accumbens was microdissected from the brain hemisphere stored in Tissue-Tek O.C.T. Compound at −70 °C. To visualize brain structures, preliminary sections were Nissl-stained (0.5%). The nucleus accumbens region was then identified according to the Allen Mouse Brain Atlas coordinates: from 2.35 mm to 0.475 mm relative to bregma (corresponding to levels 9–18 in the Allen Mouse Brain Atlas sagittal projection), with a target area of 1 × 1 mm^2^.

Six 400-μm coronal sections were prepared from each hemisphere using a Microm HM 550 cryostat (MICROM, Walldorf, Germany) and mounted on glass slides. Using a Sample Corer biopsy punch (#18035-01, Fine Science Tools, Foster City, CA, USA) with a 1 mm internal diameter, nucleus accumbens tissue from each animal was collected into a single tube containing ExtractRNA reagent (Evrogen, Moscow, Russia) and stored at −70 °C until RNA extraction. All procedures were performed at −20 °C within the cryostat chamber.

### 4.9. RNA Extraction and Real-Time PCR

RNA was subsequently extracted from the frozen tissue samples using ExtractRNA reagent (Evrogen, Moscow, Russia) according to the manufacturer’s protocol. The samples were treated with DNase I (New England Biolabs, Ipswich, MA, USA) and purified using Vazyme VAHTS RNA Clean Beads (Vazyme, Nanjing, China). RNA quality and concentration were measured with a NanoDrop 2000 spectrophotometer (Thermo Fisher Scientific, Hampton, NH, USA).

Complementary DNA was synthesized using the RevertAid kit (Thermo Fisher Scientific, Hampton, NH, USA). Reverse transcription was performed with 0.5 μg of RNA per reaction, with all procedures following manufacturer’s protocols.

Gene expression was analyzed by real-time PCR on a CFX96 thermocycler (Bio-Rad, Hercules, CA, USA) using either SYBR Green intercalating dye (for *Th*, *Maoa*, *Ppp1r1b*, *Slc6a3*) or fluorescently labeled probes (for the remaining genes). PCR was performed using Biomaster HS-qPCR SYBR Blue and Biomaster HS qPCR kits (BioLabMix, Novosibirsk, Russia), respectively, with the following amplification protocol: 95 °C for 5 min, followed by 40 cycles of denaturation at 95 °C for 10 s, and combined annealing/extension at 60 °C for 20 s with fluorescence detection.

We quantified expression of the following genes:

Hypothalamus: *Aif1*, *Il1b*, *Cxcl10*, *Traf6*, *Trem2*, *Gfap*, *Mrc1*, *Itgam*, *Crh*, *Crhr1*, *Crhbp*, *Nr3c1*, *Fkbp5*, *Abcb1a*, *Drd1*, *Drd2*, *Th*, *Maoa*, *Ppp1r1b*, *Slc6a3*.

Nucleus accumbens: *Aif1*, *Cxcl10*, *Trem2*, *Gfap*, *Nr3c1*, *Fkbp5*, *Drd1*, *Drd2*, *Drd3*.

Primer and probe sequences were designed using PrimerBlast and custom-synthesized by Biosset (Novosibirsk, Russia) and DNA-Synthesis (Moscow, Russia), respectively. PCR results were normalized to the reference genes *Hk1* (Hexokinase I) and *Pik3c3* (Phosphatidylinositol 3-kinase catalytic subunit type 3) and analyzed using the ΔΔCt method. Reference gene stability was verified in CFX Manager v. 3.1 software (coefficient of variation < 0.25, expression stability M-value < 0.5). All reactions were performed in duplicate. Primer and probe sequences are provided in Appendix A.

### 4.10. Data Analysis

Video footage was analyzed using BORIS v. 8.20.4 software [97] to manually label timings of the experimental paradigm phases as well as to tag animal behaviors (poking out). Frequency and duration of behaviors during the entire test were scored by a well-trained observer. Locomotor activity in the “Pathological aggression” test was automatically tracked using Noldus EthoVision XT 17.5 (Noldus Information Technology, Wageningen, The Netherlands), measuring the duration of active movements lasting more than 0.5 s. The personnel analyzing the data were blind to the group assignments.

A Shapiro–Wilk test was used to check whether the quantification results were normally distributed. Statistical analysis for group and cluster comparisons was performed using two-way ANOVA (factors: “aggression” [Aggressors and Control] or “cluster” [LA, NA, PA, and Control] and “treatment” [Saline and LPS]) with Tukey HSD post hoc testing. For comparisons of behavioral data across time points P1, P2, and P3, repeated measures ANOVA with Tukey HSD post hoc analysis was applied. Correlation analysis was conducted using Pearson correlation. Differences between experimental groups were considered statistically significant at *p* < 0.05, while trends were noted at *p* < 0.1.

Cluster analysis was performed using k-means clustering with k = 3 and 10 iterations, based on parameters from the “Pathological Aggression” and “Aggressive Confrontation with a Partner” tests (latency to first attack and total attack time). Data analysis was conducted using Statistica 8.0 software. All data are presented as mean ± SEM.

## Figures and Tables

**Figure 1 ijms-26-12007-f001:**
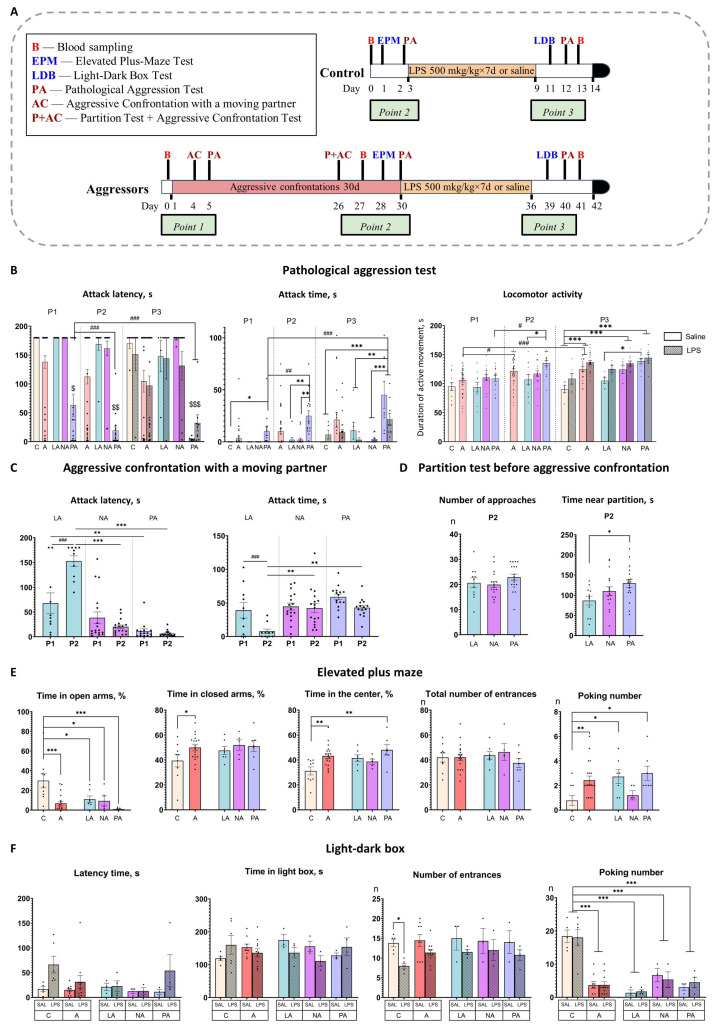
Behavioral tests results. (**A**)—Experimental design. Mice were exposed to aggression for 30 days, then to LPS/Saline injections for 7 days. Control mice were only exposed to LPS/Saline injections. B—blood sampling from retro-orbital sinus for complete blood count; EPM—Elevated plus maze test; LDB—Light-dark box test; PA—“Pathological aggression” test; P—“Partition test”; AC—“Aggressive confrontation with a moving partner” test; Point 1—time point at the beginning of aggressive confrontations; Point 2—time point after 30 days of aggressive confrontations; Point 3—time point after chronic inflammation; red box—aggressive confrontations; yellow box—chronic inflammation; black box-brain and organ isolation. (**B**)—“Pathological aggression” test; (**C**)—“Aggressive confrontation with a moving partner” test; (**D**)—“Partition test”; (**E**)—Elevated plus maze test; (**F**)—Light-dark box test. P1—beginning of aggressive confrontations; P2—the end of aggressive confrontations; P3—after fighting deprivation/chronic inflammation; C—control mice without aggressive experience; A—all mice with 30-day aggressive experience; LA—Low Aggression mice; NA—Non-pathological Aggressors; PA—Pathological Aggressors; Saline—7-day saline treatment; LPS—7-day LPS treatment; Total number of entrances in elevated plus-maze test–combined number of entrances to open arms and closed arms; Number of entrances in Light-dark test– number of entrances in the light box. * *p* < 0.05, ** *p* < 0.01, *** *p* < 0.001 compared to Control (Control vs. Aggressors, Control vs. LA vs. NA vs. PA), two-way ANOVA with Tukey HSD post hoc test. # *p* < 0.05, ## *p* < 0.01, ### *p* < 0.001 comparison between different time points (P1 vs. P2 vs. P3), $ *p* < 0.01, $$ *p* < 0.001, $$$ *p* < 0.001 comparison of PA vs. other groups, repeated measures ANOVA with Tukey HSD post hoc test.

**Figure 2 ijms-26-12007-f002:**
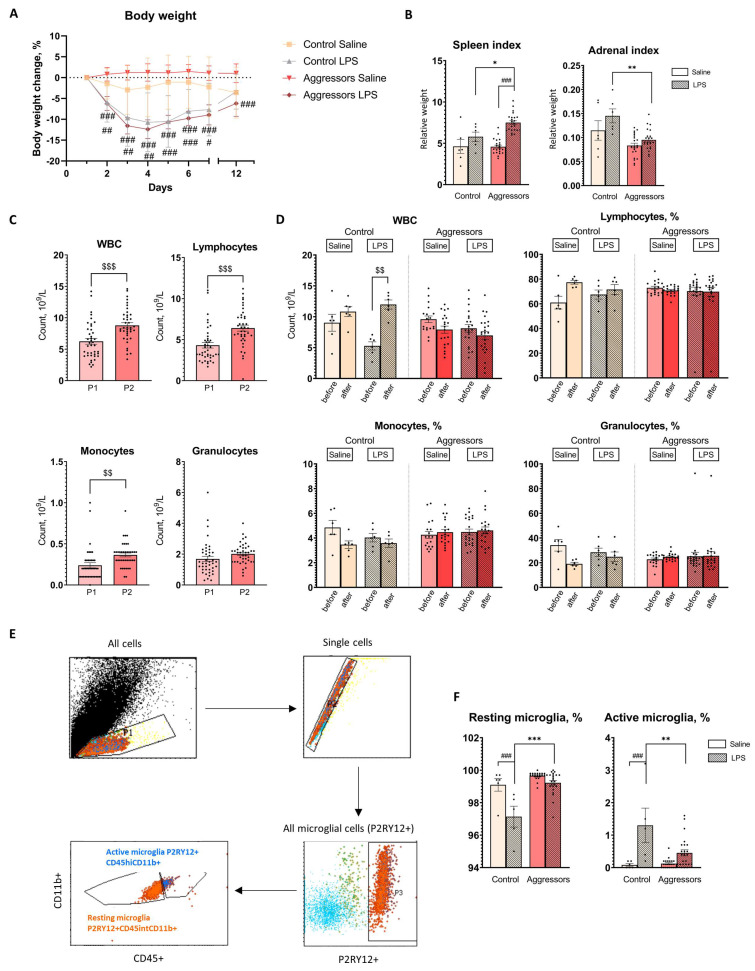
Response to LPS treatment. (**A**). Percentage body weight changes for each treatment group during LPS treatment and after 5 days; (**B**). Organ indexes (organ weight (g)/body weight (g)); (**C**). Complete blood count changes in mice before (P1) and after prolonged experience of aggression (P2); (**D**). Complete blood count changes after 7-day LPS treatment; (**E**). Gating strategy of microglial cells; (**F**). Changes in microglial population 5 days after LPS treatment. Control—mice without aggressive experience; Aggressors—mice with 30-day aggressive experience; Saline—7-day saline treatment; LPS—7-day LPS treatment; P1—beginning of aggressive confrontations; P2—the end of aggressive confrontations; before—CBC before LPS/Saline treatment; after—CBC after LPS/Saline treatment; CD45+—immune cells marker; CD11b+—macrophage cells marker; P2RY12+—microglial cells marker; P2RY12+CD45hiCD11b+—population of activated microglia; P2RY12 + CD45intCD11b+—population of resting microglia; * *p* < 0.05, ** *p* < 0.01, *** *p* < 0.001 Aggressors vs. Control; # *p* < 0.05, ## *p* < 0.01, ### *p* < 0.001 LPS vs. Saline, two-way ANOVA with Tukey HSD post hoc test; $$ *p* < 0.01, $$$ *p* < 0.001 between time points (P1 vs. P2), repeated measures ANOVA with Tukey HSD post hoc test.

**Figure 3 ijms-26-12007-f003:**
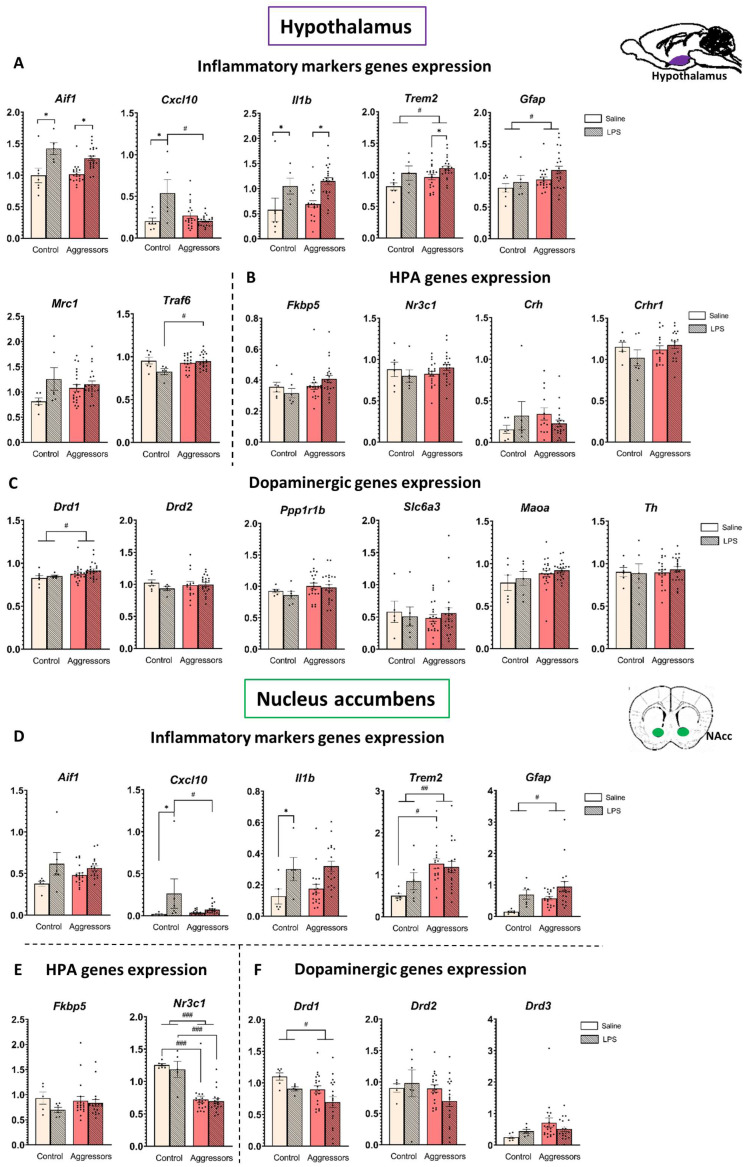
Gene expression in the hypothalamus and nucleus accumbens. (**A**). Inflammatory markers genes expression in hypothalamus; (**B**). HPA genes expression in hypothalamus; (**C**). Dopaminergic genes expression in hypothalamus; (**D**). Inflammatory markers genes expression in NAc; (**E**). HPA genes expression in NAc; (**F**). Dopaminergic genes expression in NAc. Control—mice without aggressive experience; Aggressors—mice with 30-day aggressive experience; Saline—5-day saline treatment; LPS—5-day LPS treatment; Y-Axis—expression normalized on reference genes. * *p* < 0.05, LPS vs. Saline. # *p* < 0.05, ## *p* < 0.01, ### *p* < 0.001 Aggressors vs. Control; two-way ANOVA with Tukey HSD post hoc test.

**Figure 4 ijms-26-12007-f004:**
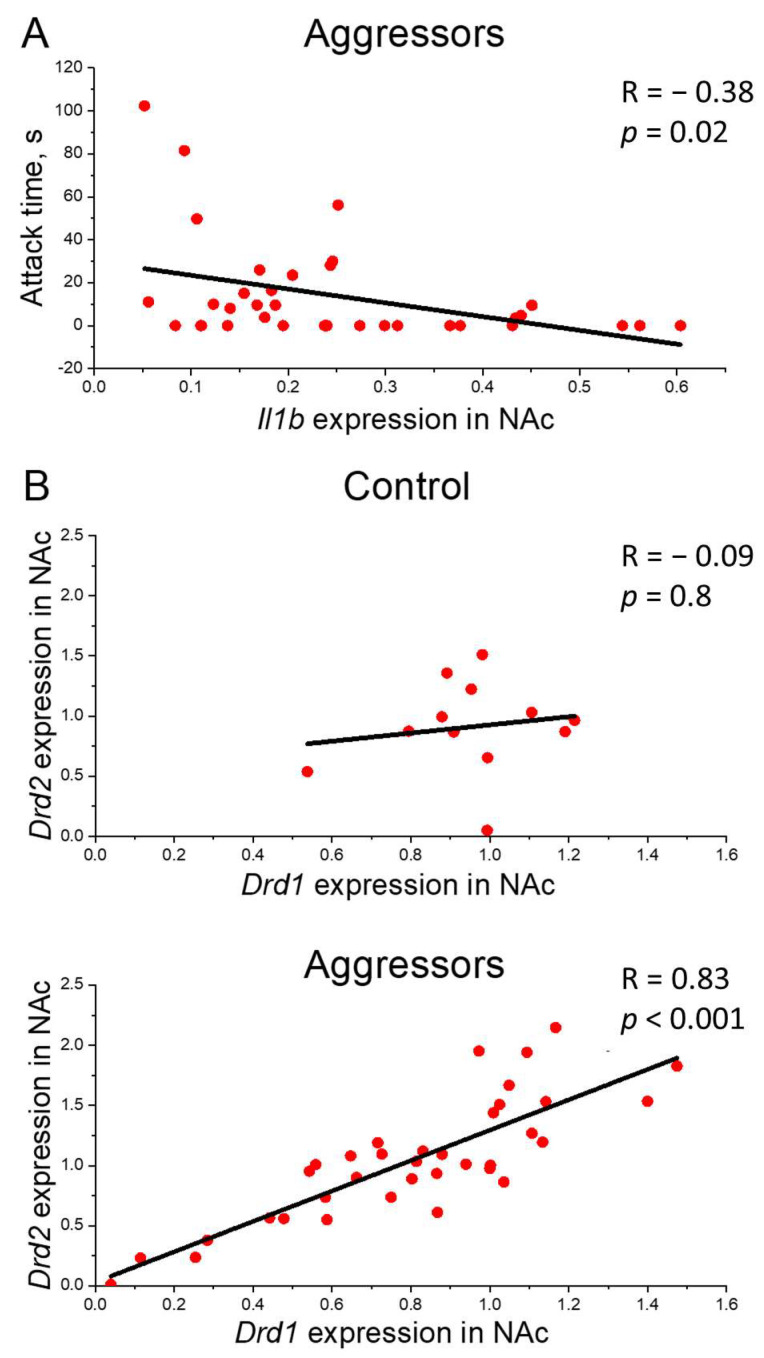
Correlation analysis. (**A**)—Correlation of pathological aggression and *Il1b* gene expression. Attack time, s—Attack time in Pathological aggression test in P3, *Il1b* gene expression in NAc—*Il1b* gene expression in nucleus accumbens normalized on reference genes; (**B**)—Correlation between *Drd1* and *Drd2* gene expression. *Drd1*, *Drd2* gene expression in NAc—*Drd1*, *Drd2* gene expression in nucleus accumbens normalized on reference genes; Control—mice without aggressive experience; Aggressors—mice with 30-day aggressive experience; R—Pearson correlation coefficient, *p*—*p* value; Pearson correlation test.

## Data Availability

The original contributions presented in this study are included in the article/Appendix A. Further inquiries can be directed to the corresponding author.

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
