# Peer review of "Prolonged Aggressive Experience Accelerates Resolution of Inflammation in Blood and Microglia After Repeated LPS Treatment"

_ijms, 2025, doi:10.3390/ijms262412007_

Round 1
Reviewer 1 Report
Comments and Suggestions for Authors
The Manuscript “Prolonged aggressive experience accelerates resolution of inflammation in blood and microglia after repeated LPS treatment” addresses very interesting topic and presents a lot of experimental results and it would be a useful addition to the literature; however, some issues should be addressed. The relevant points are outlined in detail in the text below.
Why tests for anxiety (EPM, LDB) were not performed at P1 time point, before aggression confrontation to capture baseline anxiety behavior? Why different anxiety tests were used in two time points (P2 and P3); why the Authors did not use only EPM or LDB or both in both time points to compare the same parameters before and after LPS treatment?
Anesthetized CD1 male mice were used in Pathological aggression test. Why anesthetized C57Bl/6 mice, which were used as submissive in the experimental model, were not used in this test as well?
Why tests for measuring aggressive behavior (AC and P+AC) were not done at P3 time point when the resolution of aggressive behavior is important parameter monitored in this study?
Partition test – were the C57Bl/6 mice removed from the cage for some time prior to the beginning of the test, or were they continuously housed together with the CD1? In the latter case, to what extent does it make sense to monitor their behavior once they have already become accustomed to the presence of C57Bl/6 mice?
Fig1. E and F – The number of entrances – should be specified whether the entrances in open armes/light box or closed arms/dark box.
Line 94 - ”Low Aggression mice” (LA, 24% of mice); line 96 - “Non-pathological Aggressors” (NА, 40% of mice), line 99 - “Pathological Aggressors” (PА, 36% of mice): the number of animals in each group should be stated
Line 131-133 “Analysis of aggressors as a whole (combining all subgroups) showed that they displayed a significant increase in locomotor activity throughout the "Pathological aggression" test compared to control mice” – significant difference in locomotor activity between control and whole aggressors was not shown in Figure1 B so this statement should be checked and corrected if necessary.
Line 165-169 “Thus, two days after the last LPS treatment, control animals still exhibited some features of sickness behavior: increased latency and a reduced number of entrances to the light compartment. Aggressive mice reduced the number of peeks from the dark box, indicating a decreased risk assessment [43], and did not exhibit signs of sickness behavior, suggesting that the inflammatory response had already ended in the aggressors by this time.” - Latency and a number of entrances into the light compartment of LDB are the only parameters based on what the authors concluded that inflammatory response had ended in the aggressors, but not in controls, after two days after the last LPS treatment. This statement would be more grounded if other tests for sickness behavior were used like open field test, food and water intake monitoring, home cage activity monitoring… The Authors should explain why they decided to use only LDB to test sickness behavior since the differences in duration of sickness behavior between controls and aggressors are the backbone of this study?
Line 172-174 “Chronic LPS treatment fails to reduce aggression in all subgroups. Moreover, NA and PA groups, which received LPS, demonstrated significantly higher locomotor activity compared to a control group that received LPS (p = 174 0.04 and p = 0.002, respectively; Figure 1B).” – LPS treatment in PA group increased attack latency and decreased attack time (trend is quite pronounced visually, yet no statistical significance is indicated). I kindly ask the Authors to re‑examine the analysis and confirm whether there is indeed no statistical significance, or to provide additional clarification in the figure.
Line 203: „The relative spleen weight was influenced by LPS treatment [F(2,48) = 28.6, p < 0.001], aggression [F(2,48) = 4.77, p < 0.033]...“ Spleen weights in controls and aggressors are quite the same judging by the Figure 2B. I kindly ask the Authors to re‑examine the analysis and confirm whether relative spleen weight was influenced by aggression.
Line 207: „Aggression experience significantly decreased the relative weight of the adrenal glands compared to the control [F(1,56) = 20.8, p < 0.001].“ No statistical significance is indicated in the case of adrenal glands weight in aggressors compared to controls in Figure 2B. I kindly ask the Authors to re‑examine the analysis and provide additional clarification in the Figure 2B if needed.
Lines 218-220: „After LPS induced inflammation (P3), aggressive experience affected WBC [F(2,48) = 20.7, p < 0.001] and lymphocyte percentage [F(2,48) = 4.29, p = 0.044].“ I kindly ask the Authors to re‑examine their statement that aggressive experience affected lymphocyte percentage since no statistical significance in lymphocyte percentage was found among experimental groups judging by the Figure 2D.
Line 222-224: „Thus, prolonged experience of aggression led to an increase in leukocytes, in particular lymphocytes, in the blood of mice and a more rapid resolution of LPS-induced inflammation.“ I kindly ask the Authors to explain based on what, at this point, they concluded that aggression led to a more rapid resolution of LPS-induced inflammation.
Line 254-256: „Consistent with results in the peripheral immune system, in the brain aggressors demonstrate faster resolution of LPS-induced inflammation.“ The question arises as to whether the aggressive experience accelerated the resolution of inflammation in the periphery and in the brain of LPS-treated animals, or whether the immune response in aggressive animals was simply of reduced intensity from the outset compared to controls. Do the Authors have any results or arguments that could explain why the findings of this study support a faster resolution of inflammation in aggressive animals rather than a diminished immune response to LPS treatment, given that inflammatory parameters were not measured immediately after the completion of LPS administration?
Line 478: „The exception was Il1b gene expression in the nucleus accumbens, which positively correlated with pathological aggression levels.“ No statistical significance in IL1b level in nucleus accumbes among different aggressor groups (LA, NA, PA) was indicated in Figure S2. Thus, I kindly ask the Authors to re‑examine this statement. As well as the statement „Notably, while a subset of mice developed maladaptive pathological aggression, the neuroimmune profile showed minimal correlation with behavioral subgroups except for NAc Il1b expression levels“ (line 566).
Line 608 – What tests were used to test sickness behavior?
Line 697 „Microglia were isolated using a modified protocol based on [96].“ This sentence lacks the ending.
Since this aggression model imposes stress to both submissive rats and aggressors, and that corticosterone may promote inflammation but also exert anti-inflammatory effects in immune regulation, it would be very informative to study corticosterone levels at P1, P2 and P3 time points (preposition for future studies).
Author Response
Thank you for your comments, which have significantly improved the article. We sincerely appreciate the thoughtful and constructive comments provided. We have carefully considered all points and have made revisions to the manuscript accordingly. We believe these changes have substantially improved the clarity and impact of our work.
Below, we provide a point-by-point response to your comments. All changes in the revised manuscript are highlighted in red for your convenience.
Comment 1: Why tests for anxiety (EPM, LDB) were not performed at P1 time point, before aggression confrontation to capture baseline anxiety behavior? Why different anxiety tests were used in two time points (P2 and P3); why the Authors did not use only EPM or LDB or both in both time points to compare the same parameters before and after LPS treatment?
Response 1: Thank you for your comment. We agree that while using one test allows us to compare the same parameters, the conventional approach employs a single type of test at a single time. Repeating the same behavioral test on the same animals causes carryover effects and reduces anxiety. Using the identical test at all three timepoints would simplify comparison; however, the first test can affect the animal's behavior in the second due to learning or habituation. To mitigate negative effects and improve reliability, we used EPM test before LPS treatment and LDB test after LPS treatment. Changes in anxiety levels were assessed relative to the control group at each time point. This strategy minimizes the impact of prior experience while allowing for a more robust assessment across time.
Comment 2: Anesthetized CD1 male mice were used in the Pathological aggression test. Why anesthetized C57Bl/6 mice, which were used as submissive in the experimental model, were not used in this test as well?
Response 2: Thank you for your comment, we agree that our experimental approach may lack clarity in this situation. Since aggressors had a long experience of aggressive confrontation against the moving C57Bl/6 mouse, it could propel them to react aggressively to an anesthetized C57Bl/6 mouse. To determine whether the observed aggression was indeed abnormal, we decided to use anesthetized conspecifics as a more appropriate stimulus. We selected anesthetized CD1 male mice that were matched in weight and age to the experimental mice. These mice are larger than C57BL/6 mice and represent a novel, unfamiliar stimulus. In previous experiments we tried to conduct tests using mobile juvenile CD1 males, which normally also should not elicit any aggression. However, since the aggressors displayed intense aggression toward the juveniles, we determined that using anesthetized conspecifics was a more humane experimental approach.
Comment 3: Why tests for measuring aggressive behavior (AC and P+AC) were not done at P3 time point when the resolution of aggressive behavior is important parameter monitored in this study?
Response 3: Thank you for your question. In our previous study [Mutovina et al., 2025 - preprint] we established a model to determine differential groups of aggressors (LA, NA, PA). This model requires a period of “deprivation” from aggression to successfully group animals. At the P3 time point mice have been in the deprivation from aggression stage of the experiment (there haven’t been any daily fights in the duration of LPS treatment). To maintain this state, we could not conduct the AC and AC+P tests. But at the end of the experiment, we conducted a Pathological aggression test to assess the development of aggressive behavior, which we considered sufficient to determine the specific type of aggressive behavior.
Mutovina, A.; Sapronova, A.A.; Mezhevalova, P.S.; Airiyants, K.A.; Ryabushkina, Y.A.; Salman, R.; Bondar, N.P. “The development of pathological aggression in mice during prolonged experience of aggression: behavioral and molecular changes”. Research gate 2025, [preprint] DOI: 10.13140/RG.2.2.23334.59205 Manuscript accepted for publication in Zh. Vyssh. Nerv. Deiat. Im. I. P. Pavlova, 2026 (in Russian)
Comment 4: Partition test – were the C57Bl/6 mice removed from the cage for some time prior to the beginning of the test, or were they continuously housed together with the CD1? In the latter case, to what extent does it make sense to monitor their behavior once they have already become accustomed to the presence of C57Bl/6 mice?
Response 4: Thank you for your comment. The Partition test allows for monitoring social exploratory behavior with different types of partners, and the interpretation of the results obtained depends on this. In our experiment we use the Partition test to monitor the aggressor’s behavior in the home cage with a familiar partner, and the C57Bl/6 mice were not removed from the cage before the testing, because for aggressors in the sensory contact model, the Partition test measures aggressive motivation and not the social exploratory behavior. Previous studies demonstrate that in the sensory contact model, aggressors spent a lot of time close to the partition, while losing mice avoided approaching the partition (Kudryavtseva, 2003).
We added clarification in Results (page 4, lines 141-143) and Methods (page 19, lines 718, 720):
"2.1. Prolonged experience of aggression leads to the development of three distinct patterns of ag-gressive behavior
Aggressive motivation was assessed using the “Partition test” in the home cage with a familiar partner immediately before the confrontation at P2. It is measured in the time aggressors spend close to the partition [40]. Significant differences were observed between subgroups, with the PA group spending significantly more time near the partition than the LA group (p<0.05; Figure 1D). This elevated motivation in PA mice was accompanied by increased locomotor activity in the "Pathological Aggression" test."
"4.5.3. Partition test
To assess aggressive motivation toward a familiar cage partner, we quantified behavioral parameters near the partition separating the animals [40]. The test was conducted in the experimental cage at P2 immediately prior to the "Aggressive confrontation with a partner" test. During the 5-minute "Partition Test," mouse behavior was recorded before the partition was opened for aggressive confrontations."
Kudryavtseva NN. Use of the "partition" test in behavioral and pharmacological experiments. Neurosci Behav Physiol. 2003 Jun;33(5):461-71. doi: 10.1023/a:1023411217051.
Comment 5: Fig1. E and F – The number of entrances – should be specified whether the entrances in open armes/light box or closed arms/dark box.
Response 5: Thank you for pointing it out. We specified this information in the figure description (page 6, paragraph 1, lines 198-200). Added description: “Total number of entrances in elevated plus-maze test – combined number of entrances to open arms and closed arms; Number of entrances in Light-dark test– number of entrances in the light box.”
Comment 6: Line 94 - ”Low Aggression mice” (LA, 24% of mice); line 96 - “Non-pathological Aggressors” (NА, 40% of mice), line 99 - “Pathological Aggressors” (PА, 36% of mice): the number of animals in each group should be stated
Response 6: We agree with the suggestion, and we added the number of mice (page 3, lines 95, 97, 100). Added clarification: (LA, 24% of mice (N=11), NА, 40% of mice (N=18), PА, 36% of mice (N=17).)
Comment 7: Line 131-133 “Analysis of aggressors as a whole (combining all subgroups) showed that they displayed a significant increase in locomotor activity throughout the "Pathological aggression" test compared to control mice” – significant difference in locomotor activity between control and whole aggressors was not shown in Figure 1 B so this statement should be checked and corrected if necessary.
Response 7: Thank you for your comment. A difference between control and whole aggressors groups is significant only in P3 which is shown in Figure 1B Locomotor activity at P3. With the sentence you cited, we were trying to convey that, unlike the control group, the whole aggressor’s group significantly increases locomotor activity between time points P1, P2, P3. Considering your comment, we rewrote that paragraph: “Analysis of aggressors as a whole (combining all subgroups) showed that they displayed a significant increase in locomotor activity in the "Pathological aggression" test between time points, unlike control mice, which did not change their locomotor activity. In the whole aggressors, locomotor activity increased between points P1-P2 (p = 0.0012), P2-P3 (p = 0.006), and P1-P3 (p < 0.001, Figure 1B). Also, NA and PA saline-treated groups, as well as whole aggressors, demonstrated significantly higher locomotor activity compared to a saline-treated control at P3 (NA p = 0.008, PA p < 0.001, whole aggressors p < 0.001; Figure 1B).” (Page 4, lines 133-140)
Comment 8: Line 165-169 “Thus, two days after the last LPS treatment, control animals still exhibited some features of sickness behavior: increased latency and a reduced number of entrances to the light compartment. Aggressive mice reduced the number of peeks from the dark box, indicating a decreased risk assessment [43], and did not exhibit signs of sickness behavior, suggesting that the inflammatory response had already ended in the aggressors by this time.” - Latency and a number of entrances into the light compartment of LDB are the only parameters based on what the authors concluded that inflammatory response had ended in the aggressors, but not in controls, after two days after the last LPS treatment. This statement would be more grounded if other tests for sickness behavior were used like open field test, food and water intake monitoring, home cage activity monitoring… The Authors should explain why they decided to use only LDB to test sickness behavior since the differences in duration of sickness behavior between controls and aggressors are the backbone of this study?
Response 8: Thank you for pointing it out. While the Light-dark box (LDB) test was employed to assess anxiety, our observations suggest it may also reflect a broader sickness-like state. Although we did not conduct specific tests for sickness behavior, we noted that the mice's latency and frequency of entries into the light compartment could be linked to their overall condition. Therefore, we added a more cautious attempt to interpret these LDB measures as potential indirect indicators of sickness behavior (page 4, lines 172-176):
“Thus, two days after the last LPS treatment, control animals exhibit increased latency and a reduced number of entrances to the light compartment, which could indicate a decrease in locomotion. We hypothesize that parameters may be indirect markers of sickness behavior. Aggressive mice reduced the number of peeks from the dark box, indicating a decreased risk assessment [43], and did not exhibit any indirect signs of sickness behavior, suggesting the possibility that the inflammatory response had already ended in the aggressors by this time. “
Comment 9: Line 172-174 “Chronic LPS treatment fails to reduce aggression in all subgroups. Moreover, NA and PA groups, which received LPS, demonstrated significantly higher locomotor activity compared to a control group that received LPS (p = 174 0.04 and p = 0.002, respectively; Figure 1B).” – LPS treatment in PA group increased attack latency and decreased attack time (trend is quite pronounced visually, yet no statistical significance is indicated). I kindly ask the Authors to re‑examine the analysis and confirm whether there is indeed no statistical significance, or to provide additional clarification in the figure.
Response 9: Thank you for pointing this out. There is indeed a noticeable difference between the means of each group, but because of the wide range of data, no statistical significance was found for latency (F = 1.336, p = 0.253, for PA Saline min = 0.95, max = 10.75, Std Dev = 3.07, Std Err = 1.024; for PA LPS min = 2.24, max = 138.15, Std Dev = 41.94, Std Err = 13.98) or for attack time (F = 1.16, p = 0.284, for PA Saline min = 8.08, max = 102.35, Std Dev = 36.47, Std Err = 12.15; for PA LPS min = 4.75, max = 56.13, Std Dev = 17.69, Std Err = 5.89). We also tried to apply nonparametric statistical methods, but they also did not reveal significant differences.
Comment 10: Line 203: „The relative spleen weight was influenced by LPS treatment [F(2,48) = 28.6, p < 0.001], aggression [F(2,48) = 4.77, p < 0.033]...“ Spleen weights in controls and aggressors are quite the same judging by the Figure 2B. I kindly ask the Authors to re‑examine the analysis and confirm whether relative spleen weight was influenced by aggression.
Response 10: Thank you for your comment. For these parameters we found factor effects, but additional pairwise comparisons between groups did not find any significant differences. A "factor effect" (more formally called a "main effect") refers to the impact of a single independent variable (or "factor") on a dependent variable. It is the overall difference in the mean of the dependent variable(s) across the different levels of that single factor, averaged across the levels of any other factors in the study. ANOVA partitions the total variability in the data into the variance explained by the main effects, the variance explained by the interaction effects, and the remaining unexplained error variance. The results of the ANOVA, typically presented as F-statistics and p-values, indicate whether these effects are statistically significant. Multiple comparisons, on the other hand, refer to a set of statistical tests performed after an initial analysis (like ANOVA) shows an overall significant difference among group means. Their purpose is to pinpoint exactly which specific pairs of group means are statistically different from each other. The key difference is that factor analysis deals with hidden variables and correlations between variables, while multiple comparisons deal with mean differences between specific groups.
To clear that misunderstanding, we added the phrase that post-hoc comparisons were not significant. (page 6, lines 213-216), and more clearly indicated when we talk about factor effects. “The relative spleen weight was influenced by factors: “LPS treatment” [F(2,48) = 28.6, p < 0.001], “aggression” [F(2,48) = 4.77, p < 0.033], and the interaction between “LPS treatment” and “aggression” [F(2,48) = 5.21, p < 0.026], but post-hoc analysis did not find any significant differences between aggressors and control with different treatment. LPS treatment resulted in spleen enlargement in aggressors (p < 0.001) but not in controls (Figure 2B).”
Comment 11: Line 207: „Aggression experience significantly decreased the relative weight of the adrenal glands compared to the control [F(1,56) = 20.8, p < 0.001].“ No statistical significance is indicated in the case of adrenal glands weight in aggressors compared to controls in Figure 2B. I kindly ask the Authors to re‑examine the analysis and provide additional clarification in the Figure 2B if needed.
Response 11: Thank you for pointing it out. We agree that the cited statement may be misleading. For these parameters we found factor effects, but additional pairwise comparisons did not find any significant differences. To clear that misunderstanding, we added the phrase that post-hoc comparisons were not significant, but factor effects were. (page 6, lines 219-221) “The “aggression” factor was significantly affecting the relative weight of the adrenal glands compared to the control [F(1,56) = 20.8, p < 0.001], but pair-wise comparisons were not significant. The “aggression” factor also affected adrenal hyperplasia induced by the LPS [F(1,56) = 5.6, p < 0.05]: the weight of the adrenal glands was significantly increased in the control group with LPS treatment compared to the aggressors (Figure 2B). Subgroups of aggressors did not differ in these parameters.”
Comment 12: Lines 218-220: „After LPS induced inflammation (P3), aggressive experience affected WBC [F(2,48) = 20.7, p < 0.001] and lymphocyte percentage [F(2,48) = 4.29, p = 0.044].“ I kindly ask the Authors to re‑examine their statement that aggressive experience affected lymphocyte percentage since no statistical significance in lymphocyte percentage was found among experimental groups judging by the Figure 2D.
Response 12: Thank you for this comment, we agree that we failed to clearly convey the meaning. For these parameters we found factor effects, but additional post-hoc pairwise comparisons did not find any significant differences. To clear up that misunderstanding, we added the additional phrase that post-hoc comparisons were not significant. (page 6-7, lines 234-237) “After LPS-induced inflammation (P3), factor ‘aggression’ affected WBC [F(2,48) = 20.7, p < 0.001] and lymphocyte percentage [F(2,48) = 4.29, p = 0.044], although for lymphocyte percentage no significant differences between aggressors and control with different treatment were found (according to Tukey HSD post-hoc test). “
Comment 13: Line 222-224: „Thus, prolonged experience of aggression led to an increase in leukocytes, in particular lymphocytes, in the blood of mice and a more rapid resolution of LPS-induced inflammation.“ I kindly ask the Authors to explain based on what, at this point, they concluded that aggression led to a more rapid resolution of LPS-induced inflammation.
Response 13: Thank you for pointing it out. We agree that by this point in the Results section this statement seems baseless, and we removed it from the text (page 7, lines 241-242). Nevertheless, we expand on this hypothesis more in the Discussion. We believe it is more probable that aggression led to a more rapid resolution of LPS-induced inflammation, considering our data and other research of the immune system in aggressors.
Comment 14: Line 254-256: „Consistent with results in the peripheral immune system, in the brain aggressors demonstrate faster resolution of LPS-induced inflammation.“ The question arises as to whether the aggressive experience accelerated the resolution of inflammation in the periphery and in the brain of LPS-treated animals, or whether the immune response in aggressive animals was simply of reduced intensity from the outset compared to controls. Do the Authors have any results or arguments that could explain why the findings of this study support a faster resolution of inflammation in aggressive animals rather than a diminished immune response to LPS treatment, given that inflammatory parameters were not measured immediately after the completion of LPS administration?
Response 14: Thank you for your comment. We agree that our work does not definitively confirm our hypothesis. Our assumption is based on the existing data from other researchers, which indicates a more active immune system in aggressive animals. Previous research provides compelling evidence for a hyperactive immune system in aggressive animals. A different immune response between aggressive and non-aggressive mice was also previously demonstrated by the Idova group [Alperina et al., 2019; Idova et al., 2016; Devoino et al., 1998; Idova et al., 2000], Granger [Granger et al., 1997], and Kudryavtseva [Kudryavtseva et al., 2007; Devoino et al., 1993]. Experience of aggression leads to a stronger activation of T-cell immunity: aggressors showed a significantly higher number of active T-cells in the spleen and bone marrow compared to controls after immunization [Devoino et al., 1998; Idova et al., 2000; Devoino et al., 1993]. In our study, the absence of immune activation during the delayed phase of the LPS response appears paradoxical in this context. However, we interpret this finding not as a lack of initial response, but as evidence of a faster resolution of the inflammatory process. This conclusion reconciles our data with the established paradigm of enhanced immunity in aggression. To directly test this hypothesis, future work will involve a comprehensive time-course analysis of immune parameters, from the acute phase through resolution following LPS challenge. We have addressed the limitations of our study and suggested directions for future research in the concluding section of the Discussion. (page 17, lines 616-626).
It is also important to acknowledge several methodological constraints that shape the interpretation of our results and suggest priorities for subsequent research. Our findings are constrained by the measurement of inflammatory markers at the gene expression level and the lack of temporal tracking of the immune response. To build upon this work, subsequent studies should incorporate more time points, quantify cytokine and chemokine levels at the protein level in both the periphery and brain, and perform immunohistochemical analyses of microglial and astroglial morphology. These approaches would greatly increase the translational relevance of the research and provide a more comprehensive elucidation of immune system dynamics in aggressive animals. A more holistic model would also require examining the interplay with other critical neural systems implicated in aggression, notably the opioid and serotonin pathways.
- Alperina, E.; Idova, G.; Zhukova, E.; Gevorgyan, M.; Cheido, M. Cytokine Variations Within Brain Structures in Rats Selected for Differences in Aggression. Neurosci. Lett. 2019, 692, 193–198. DOI: 10.1016/j.neulet.2018.11.012
- Devoino, L.; Alperina, E.; Kudryavtseva, N.; Popova, N. Immune Responses in Male Mice with Aggressive and Submis-sive Behavior Patterns: Strain Differences. Brain Behav. Immun. 1993, 7, 91–96.
- Devoino, L.V.; Idova, G.V.; Alperina, E.L.; Cheido, M.A. Neurochemical Set-up of the Brain - An Extra-Immune Mecha-nism of Psychoneuroimmunomodulation. Vestn. Ross. Akad. Med. Nauk 1998, 9, 19–24.
- Granger, D.A.; Hood, K.E.; Ikeda, S.C.; Reed, C.L.; Block, M.L. Effects of Peripheral Immune Activation on Social Behavior and Adrenocortical Activity in Aggressive Mice: Genotype-Environment Interactions. Aggress. Behav. 1997, 23, 93–105.
- Idova, G.V.; Pavina, T.A.; Alperina, E.L.; Devoino, L.V. Influence of Submissive and Aggressive Behavior Patterns on Changes in the Number of T-Lymphocytes CD4+ and CD8+ in Bone Marrow. Immunologiya 2000, 1, 24–26.
- Idova, G.V.; Markova, E.V.; Gevorgyan, M.M.; Alperina, E.L.; Cheido, M.A. Changes in Production of Cytokines by C57Bl/6J Mouse Spleen during Aggression Provoked by Social Stress. Bull. Exp. Biol. Med. 2016, 160, 679–682. DOI: 10.1007/s10517-016-3248-y
- Kudryavtseva, N.N.; Tenditnik, M.V.; Nikolin, V.P.; Popova, N.A.; Kaledin, V.I. The Influence of Psychoemotional Status on Metastasis of Lewis Lung Carcinoma and Hepatocarcinoma-29 in Mice of C57BL/6J and CBA/Lac Strains. Exp. Oncol. 2007, 29, 35–38.
Comment 15: Line 478: „The exception was Il1b gene expression in the nucleus accumbens, which positively correlated with pathological aggression levels.“ No statistical significance in IL1b level in nucleus accumbes among different aggressor groups (LA, NA, PA) was indicated in Figure S2. Thus, I kindly ask the Authors to re‑examine this statement. As well as the statement „Notably, while a subset of mice developed maladaptive pathological aggression, the neuroimmune profile showed minimal correlation with behavioral subgroups except for NAc Il1b expression levels“ (line 566).
Response 15: Thank you for your comment. We agree that there is indeed no statistical significance in the pairwise comparisons between these groups, but there is a correlation between level of Il1b gene expression and level of pathological aggression. Pairwise comparisons are used in analyses of differences between group means, typically after a significant Analysis of Variance (ANOVA) result. Statistical significance of correlation refers to a different statistical test entirely: determining if the strength of the relationship between two continuous variables is different in one group or condition compared to another. In this case expression of the Il1b gene positively correlated with data from the Pathological aggression test at P3 (attack time). This misunderstanding could have arisen because we moved the figure with correlation analysis results to supplement (Figure S3A). We decided to add this figure to the main text as Figure 4 for better comprehension and added the reference to this figure in every instance where we mention this correlation. (page 10, lines 318, 339, page 15, line 520), the figure is on page 11, and the added description is on page 11, lines 342-348:
“Figure 4. Correlation analysis. A - Correlation of pathological aggression and Il1b gene expression. Attack time, s — Attack time in Pathological aggression test in P3, Il1b expression in NAc — Il1b gene expression in nucleus accumbens normalized on reference genes; B - Correlation between Drd1 and Drd2 gene expression. Drd1, Drd2 expression in NAc — Drd1, Drd2 gene expression in nucleus accumbens normalized on reference genes; Control - mice without aggressive experience; Aggressors - mice with 30-day aggressive experience; R — Pearson correlation coefficient, p — p value; Pearson correlation test.”
Comment 16: Line 608 – What tests were used to test sickness behavior?
Response 16: The sickness of animals was visually assessed before the start of the LPS treatment to exclude sick animals from the experiment. We did not use specific tests for this. We added this clarification to the text. (page 18, line 664-666): "Exclusion criteria included visually-assessed sickness behavior (decreased reaction to our manipulations, prolonged periods of immobility, swelling) prior to LPS administration. "
Comment 17: Line 697 „Microglia were isolated using a modified protocol based on [96].“ This sentence lacks the ending.
Response 17: Thank you for pointing it out. We changed that sentence to “Microglia were isolated using a previously published protocol with some modifications [96].” (page 19, lines 757-758).
Comment 18: Since this aggression model imposes stress to both submissive rats and aggressors, and that corticosterone may promote inflammation but also exert anti-inflammatory effects in immune regulation, it would be very informative to study corticosterone levels at P1, P2 and P3 time points (preposition for future studies).
Response 18: Thank you for your proposition; we agree that the absence of a more profound examination of the HPA axis is one of the limitations in our study. In the most recent experiment, we measured basal corticosterone to assess the responsiveness of the HPA axis of aggressors after 30-day deprivation of aggression (Mutovina at al., preprint, accepted article) but found no differences compared to the control and no differences between subgroups of aggressors (the experiment did not include LPS treatment). In future studies, we will try to include assessment of corticosterone levels in the experimental design.
Mutovina, A.; Sapronova, A.A.; Mezhevalova, P.S.; Airiyants, K.A.; Ryabushkina, Y.A.; Salman, R.; Bondar, N.P. “The development of pathological aggression in mice during prolonged experience of aggression: behavioral and molecular changes”. Research gate 2025, [preprint] DOI: 10.13140/RG.2.2.23334.59205 Manuscript accepted for publication in Zh. Vyssh. Nerv. Deiat. Im. I. P. Pavlova, 2026 (in Russian)
Reviewer 2 Report
Comments and Suggestions for Authors
Comments and Suggestions for Authors
General Comments:
This study is well-designed, data-rich, and presents novel findings. It is suitable for publication in International Journal of Molecular Sciences pending minor revisions, primarily to enhance methodological clarity, improve figure presentation, and strengthen the discussion's logical rigor.
Specific Comments by Section:
- Methodology
1) LPS Dosage and Administration Details
Lacks details on injection volume, solvent, and frequency (e.g., once daily). Please supplement.
2) Flow Cytometry for Microglia
While markers (P2RY12, CD45, CD11b) are mentioned, antibody sources, clone numbers, and validation details are absent. Please provide antibody information and consider including a gating strategy schematic or reference.
3) Gene Expression Normalization:
Reference genes (Hk1, Pik3c3) are mentioned, but stability validation data (e.g., geNorm, NormFinder) are not provided. Please include.
- Results
1) Justification for Aggression Subgrouping
Cluster analysis used only "latency to first attack" and "total attack duration." Consider if additional behavioral parameters (e.g., attack frequency, target) should be included. Justify the clustering approach.
2) Incomplete Presentation of Gene Expression Data
Some gene expression changes (e.g., Traf6, Gfap) are described but not clearly visualized in Figure3. Consider revising the figure or supplementing with additional panels.
3) Lack of In-depth Analysis of the LA Group
The LA group behaviorally differs from PA/NA, but its immune/neuroendocrine characteristics are not thoroughly discussed. Recommend additional analysis or commentary.
4) Inconsistent Abbreviations/Labels:
"Cntrl" and "Control" are used interchangeably in figures/text. Standardize to "Control."
- Discussion
1) Relationship Between Pathological Aggression and Neuroimmune Phenotype
The PA group showed no significant immune differences from NA/LA despite stronger pathological behavior. This "behavior-immune decoupling" deserves deeper discussion.
2) Interpretation of Trem2 Role is Somewhat Speculative
While increased Trem2 expression correlates with inflammation resolution, functional evidence is lacking. Acknowledge this limitation or suggest future experimental directions.
3) Mechanistic Link Between Dopamine and Inflammation Resolution
The suggestion that D1 receptors enhance anti-inflammatory signaling lacks direct evidence. Phrase more cautiously or propose future studies to test this hypothesis.
- Data Availability
Consider providing raw data or analysis scripts for clustering and gene expression to enhance reproducibility, if applicable.
Author Response
Thank you for your high rating of our article. We sincerely appreciate the thoughtful and constructive comments provided. We have carefully considered all points and have made revisions to the manuscript accordingly. We believe these changes have substantially improved the clarity and impact of our work.
Below, we provide a point-by-point response to your comments. All changes in the revised manuscript are highlighted in red for your convenience.
- Methodology
Comment 1: LPS Dosage and Administration Details
Lacks details on injection volume, solvent, and frequency (e.g., once daily). Please supplement.
Response 1: Thank you for pointing it out. We added this information in Methods (page 17,lines 658-659): “... LPS (500 μg/kg, volume – 10 μl/g, solvent – saline, Escherichia coli serotype O55:B5, Sigma, USA) or equivalent volume of saline was injected once a day between 09:00 and 10:00 AM.”
Comment 2: Flow Cytometry for Microglia
While markers (P2RY12, CD45, CD11b) are mentioned, antibody sources, clone numbers, and validation details are absent. Please provide antibody information and consider including a gating strategy schematic or reference.
Response 2: Thank you for this suggestion. The gating strategy schematic is present in Figure 2E. We added detailed information about antibodies in Methods (page 20, lines 763-766): “CD45-PE-Cy7 (1 μg/10⁶ cells, clone 30-F11, E-AB-F1136H, Elabscience, China), CD11b-PerCP (1 μg/10⁶ cells, clone M1/70, E-AB-F1081F, Elabscience, China), and P2RY12-PE (0.2 μg/10⁶ cells, clone S16007D, 848003, Biolegend, USA).”
Comment 3: Gene Expression Normalization:
Reference genes (Hk1, Pik3c3) are mentioned, but stability validation data (e.g., geNorm, NormFinder) are not provided. Please include.
Response 3: Thank you for your comment. We tested gene expression stability with the Bio-Rad CFX Manager software. In the Bio-Rad CFX Manager software, the "Target Stability Value" is a calculated metric used in the Gene Expression module to determine the stability of a chosen reference gene for qPCR data normalization. The software calculates two quality parameters for the reference genes: Coefficient of Variation (CV) of normalized reference gene relative quantities (lower CV value denotes higher stability) and M-value, measure of the reference gene expression stability (parameter from GeNorm software). Acceptable values for stably expressed reference genes for homogeneous samples are: CV < 0.25, M < 0.5 (Hellemans et al. 2007). The reference genes we used show stable expression in our experimental groups according to these indicators (page 21, lines 810-812).
Hellemans, J., Mortier, G., De Paepe, A., Speleman, F., & Vandesompele, J. (2007). qBase relative quantification framework and software for management and automated analysis of real-time quantitative PCR data. Genome biology, 8(2), R19.
- Results
Comment 4: Justification for Aggression Subgrouping
Cluster analysis used only "latency to first attack" and "total attack duration." Consider if additional behavioral parameters (e.g., attack frequency, target) should be included. Justify the clustering approach.
Response 4: Thank you for your comment. At first, to cluster aggressive mice, we used all parameters recorded in the aggressive behavior tests. However, the parameters "latency to first attack" and "total attack duration" were the ones that made the largest contribution to the differences between the groups. Excluding the remaining parameters did not lead to changes in the composition of the clusters. Therefore, in the final version, we proceeded with clustering using only these two parameters from the 2 aggression tests (clustering was performed using a total of 4 parameters).
Comment 5: Incomplete Presentation of Gene Expression Data
Some gene expression changes (e.g., Traf6, Gfap) are described but not clearly visualized in Figure 3. Consider revising the figure or supplementing with additional panels.
Response 5: Thank you for pointing it out. The expression level of the Traf6 gene was only measured in the hypothalamus; therefore, the changes are shown in Figure 3A and described in the text accordingly: “However, the expression level of Traf6 (TRAF6 in microglia activates the NF-κB transcription factor) was higher in LPS-treated aggressors (p = 0.05 Agg-LPS vs Control-LPS), compared to control” (page 10, lines 312-314). Expression levels of the GFAP gene were measured in both the hypothalamus and nucleus accumbens and are shown in Figures 3A and 3B. The text description corresponds to the depicted changes: “Notably, in mice with aggressive experience, the expression of the Gfap gene … was increased in both structures (p < 0.03 for all)” (page 10, lines 303-306).
Comment 6: Lack of In-depth Analysis of the LA Group
The LA group behaviorally differs from PA/NA, but its immune/neuroendocrine characteristics are not thoroughly discussed. Recommend additional analysis or commentary.
Response 6: Thank you for your comment. We agree that the LA group is of significant interest for investigation. It should be noted that mice from the LA group are nevertheless aggressive: in all confrontations they exhibited dominant behavior (attack or aggressive grooming) and attacked victims in cases of non-compliance; overall, during 30 days of confrontations with a partner, low-aggressive mice attacked victims in at least half of the cases. It is possible that the specific form of aggression manifestation has a lesser impact on the immune system than the very fact of its daily occurrence. In Discussion section we have added a hypothesis regarding the absence of differences in the immune response among the aggressor subgroups (page 12, lines 399-403, page 13, lines 414-421):
"Unlike other groups, "Low Aggressive mice" show a progressive decline in aggression toward a moving partner and exhibit no pathological aggression or high agitation. It is important to note that mice from the LA group nevertheless exhibited aggressive behavior. In all confrontations, they displayed dominant actions (attacks or aggressive grooming) and attacked partners if provoked. Overall, during the 30-day period of confrontations, the LA mice attacked their partners in at least half of the cases. Moreover, they share the decision-making impairment seen in the PA group."
“In this study we were unable to detect significant differences between the aggressor subgroups in terms of immune response parameters and gene expression in the hypothalamus and nucleus accumbens (Figures S1, S2, and S3). The similar changes in the parameters we assessed across all aggressors may suggest that the prolonged manifestation of aggression, regardless of its intensity level, exerts a significant influence on the immune response. A more complete understanding of the molecular basis of pathological aggression will likely require examining its interplay with other critical neural systems, notably the opioid and serotonin pathways.“
Comment 7: Inconsistent Abbreviations/Labels:
"Cntrl" and "Control" are used interchangeably in figures/text. Standardize to "Control."
Response 7: Thank you for your comment; we standardized the label "Control" in the text and left only the abbreviation “C” in Figure 1 for better clarity in the graphs.
3. Discussion
Comment 8: Relationship Between Pathological Aggression and Neuroimmune Phenotype
The PA group showed no significant immune differences from NA/LA despite stronger pathological behavior. This "behavior-immune decoupling" deserves deeper discussion.
Response 8: Yes, we agree that it would be interesting to discuss. No noticeable decrease of aggressive behavior after LPS treatment could be attributed to the behavior-immune decoupling. In studies involving chronic social stress, subordinate mice exhibited heightened physiological inflammation when challenged with an immune activator (LPS), but failed to display the expected sickness behavior [Ashley NT, Demas GE. Neuroendocrine-immune circuits, phenotypes, and interactions. Horm Behav. 2017 Jan;87:25-34. doi: 10.1016/j.yhbeh.2016.10.004.]. They prioritized socially defensive actions over recuperative ones, meaning their outward behavior was "decoupled" from their internal inflammatory state. In our case, PA group of aggressive mice demonstrated the highest levels of aggression, independent of LPS injections. But at the same time we cannot positively confirm that it is indeed a decoupling and not related to an altered immune response, since we did not test them at the height of inflammation. We added a hypothesis regarding the absence of differences in the immune response among the aggressor subgroups (page 13, lines 414-421):
"In this study we were unable to detect significant differences between the aggressor subgroups in terms of immune response parameters and gene expression in the hypothalamus and nucleus accumbens (Figures S1, S2 and S3). The similar changes in the parameters we assessed across all aggressors may suggest that the prolonged manifestation of aggression, regardless of its intensity level, exerts a significant influence on the immune response. A more complete understanding of the molecular basis of pathological aggression will likely require examining its interplay with other critical neural systems, notably the opioid and serotonin pathways."
Comment 9: While increased Trem2 expression correlates with inflammation resolution, functional evidence is lacking. Acknowledge this limitation or suggest future experimental directions.
Response 9: Thank you for the suggestion; we agree that heightened Trem2 gene expression isn’t equal to enhanced TREM2 level and its involvement in the resolution of inflammation has not been clearly proven. We added the limitation of our assumption (page 15, lines 512-514):
"Typically, microglial activation is rapidly suppressed to prevent secondary neuronal damage [71]. The microglial receptor TREM2 facilitates this negative feedback, promoting a switch to an anti-inflammatory phenotype upon binding apoptotic cell lipids or bacterial ligands [72,73]. We found that aggressive mice exhibited twofold higher Trem2 expression in the NAc compared to controls, with a similar increase in the hypothalamus. We hypothesize that heightened gene expression leads to enhanced TREM2 level and this, in turn, may contribute to the accelerated inflammatory resolution observed in these mice."
Page 16, lines 606-608:
"The observed adaptive immune response is possibly modulated by an increased Trem2 expression and dopaminergic signaling, but this preposition requires additional research. "
Comment 10: Mechanistic Link Between Dopamine and Inflammation Resolution
The suggestion that D1 receptors enhance anti-inflammatory signaling lacks direct evidence. Phrase more cautiously or propose future studies to test this hypothesis.
Response 10: We agree with your comment, and we added a proposition for future studies (page 16, lines 606-608): “The observed adaptive immune response is possibly modulated by an increased Trem2 expression and dopaminergic signaling, but this proposition requires additional research.”
4. Data Availability
Comment 11: Consider providing raw data or analysis scripts for clustering and gene expression to enhance reproducibility, if applicable.
Response 11: Thank you for this suggestion. We will provide raw data if requested by the interested party. “All relevant data are available in the supplementary materials, and raw data can be provided by request.” (page 21, lines 846-847)
Reviewer 3 Report
Comments and Suggestions for Authors
Please see the attached file.

Author Response
Thank you for your comments, which have significantly improved the article. We sincerely appreciate the thoughtful and constructive comments provided. We have carefully considered all points and have made revisions to the manuscript accordingly. We believe these changes have substantially improved the clarity and impact of our work.
Below, we provide a point-by-point response to your comments. All changes in the revised manuscript are highlighted in red for your convenience.
Comment 1: Is aggressive behavior beneficial for our health? Does it have positive implications for our health? In the first place, aggressive behavior is an essential behavior for animals to protect themselves and survive. It is reasonable that being aggressive also protects them from inflammation.
Response 1: It is hard to tell whether aggressive behavior is beneficial or not based on our study. In the first place we tried to model a pathological aggression, which by definition implies some kind of pathology. In our work the immune status of aggressive mice indeed helped in fighting LPS-induced inflammation; it was also shown that aggressive mice exhibit reduced susceptibility to tumors: aggressive mice develop smaller tumors [Amkraut, Solomon, 1972] and exhibit higher NK cell cytotoxicity [Alperina et al., 2019], as well as fewer metastases [Costa-Pinto et al., 2009]. All of this could indicate a primed hyperinflammatory phenotype in aggressors, but it is not always profitable. It is important to remember that mice research does not necessarily translate to humans. In humans in terms of overall health, including mental state, we cannot say that aggressive behavior is beneficial. Contrary to protecting against inflammation, individuals with high aggression traits or anger disorders tend to display heightened inflammatory cytokine levels. This chronic low-grade inflammation is a risk factor for numerous diseases.
- Alperina, E.; Idova, G.; Zhukova, E.; Gevorgyan, M.; Cheido, M. Cytokine Variations Within Brain Structures in Rats Selected for Differences in Aggression. Neurosci. Lett. 2019, 692, 193–198.
- Amkraut, A.; Solomon, G.F. Stress and Murine Sarcoma Virus (Moloney)-Induced Tumors. Cancer Res. 1972, 32, 1428–1433.
- Costa-Pinto, F.A.; Cohn, D.W.H.; Sa-Rocha, V.M.; Sa-Rocha, L.C.; Palermo-Neto, J. Behavior: A Relevant Tool for Brain–Immune System Interaction Studies. Ann. N. Y. Acad. Sci. 2009, 1153, 107–119.
Comment 2: It is quite interesting to know how the subordinate, that is subject to prolonged aggressive behavior of dominant animals, respond to inflammation caused by LPS.
Response 2: That is indeed quite an interesting subject, considering the research on the immune status of depressed individuals is gaining a lot of attention. And it was shown that subordinate mice develop a depression-like state. It would be interesting to address in future studies. Our group is actually in the process of working on this question.
Comment 3: In this study, the aggressive experience period is relatively long, 30 days, but will the same happen if it is shorter?
Response 3: Indeed, there are studies that use a 20-day period of aggressive experience [Smagin et al., 2010; Kudryavtseva et al., 2011] within a similar experimental paradigm. These studies demonstrate that a 20-day period may be sufficient for forming a stable aggressive behavioral type in mice and consider mice with a high level of aggression towards a partner as pathological aggressors. Our goal was to induce precisely this pathological type of aggressive behavior, which is why we used a longer period. However, we did not test whether such pathological behavior is formed within 20 days. We conducted the test for pathological aggression after 4 and 30 days of aggressive interactions and showed that during this time, the level of aggression towards an anesthetized partner doubles. It is possible that 20 days of aggressive experience might also be sufficient, but this requires separate experiments. For male CD1 mice, just 3 days of the resident-intruder paradigm is sufficient to separate animals into aggressive and non-aggressive groups and to find a correlation between the level of aggression and interleukin Il1b levels (Takahashi et al., 2022). Considering other mouse strain, for instance, the SAL strain, genetically selected for high levels of aggression, they show high aggression in the test towards an anesthetized male after only 3 days of the resident-intruder test (Natarajan et al., 2009), and exhibit aggression towards a female after 9 days of the resident-intruder paradigm (Caramaschi et al., 2008).
- Caramaschi, D., de Boer, S. F., de Vries, H., & Koolhaas, J. M. (2008). Development of violence in mice through repeated victory along with changes in prefrontal cortex neurochemistry. Behav Brain Res, 189(2), 263-272. doi: 10.1016/j.bbr.2008.01.003
- Natarajan, D., de Vries, H., Saaltink, D. J., de Boer, S. F., & Koolhaas, J. M. (2009). Delineation of violence from functional aggression in mice: an ethological approach. Behav Genet, 39(1), 73-90. doi: 10.1007/s10519-008-9230-3
- Kudryavtseva, N.N.; Smagin, D.A.; Bondar, N.P. Modeling Fighting Deprivation Effect in Mouse Repeated Aggression Paradigm. Prog. Neuropsychopharmacol. Biol. Psychiatry 2011, 35, 1472–1478.
Smagin, D.A.; Bondar, N.P.; Kudryavtseva, N.N. Repeated Experience of Aggression and Consequences of Deprivation in Male Mice. Psikhofarmakol. Biol. Narkol. 2010, 10, 2636–2648. - Takahashi, A., Aleyasin, H., Stavarache, M. A., Li, L., Cathomas, F., Parise, L. F., . . . Russo, S. J. (2022). Neuromodulatory effect of interleukin 1beta in the dorsal raphe nucleus on individual differences in aggression. Mol Psychiatry, 27(5), 2563-2579.
Comment 4. In the discussion, it is desirable to state the limitation of the study along with the direction and issues of future research.
Response 4: We added the limitations of our study and possible directions of future research in the concluding section of the Discussion (page 17, lines 616-626):
"It is also important to acknowledge several methodological constraints that shape the interpretation of our results and suggest priorities for subsequent research. Our findings are constrained by the measurement of inflammatory markers at the gene expression level and the lack of temporal tracking of the immune response. To build upon this work, subsequent studies should incorporate more time points, quantify cytokine and chemokine levels at the protein level in both the periphery and brain, and perform immunohistochemical analyses of microglial and astroglial morphology. These approaches would greatly increase the translational relevance of the research and provide a more comprehensive elucidation of immune system dynamics in aggressive animals. A more holistic model would also require examining the interplay with other critical neural systems implicated in aggression, notably the opioid and serotonin pathways."
Comment 5: Minor points: Line 152; “increased poking from the closed arms number” should be “increased number of poking from the closed arms”
Response 5: Thank you for your suggestion; we changed that phrase in the text (page 4, line 157): “Aggression experience also led to increased number of poking from the closed arms “
Comment 6: Lines 707-708; From live single cells, the next gate is cells positive for P2RY12+ (the average microglial (P2RY12+) cell yield from half of a mouse brain was 40000 cells) This sentence is quite hard to understand.
Response 6: Thank you for your suggestion, we have corrected the phrase in the manuscript, breaking it into two sentences. “From live single cells, the next gate is cells positive for P2RY12+. The next gates were determined by the CD45+/CD11b+ positivity and differentiated by high and low expression of CD45+: resting microglia P2RY12+/CD45intermediate/CD11b+ and active microglia P2RY12+/CD45high/CD11b+ (Figure 2E).” (page 20, line 768-772).
Round 2
Reviewer 1 Report
Comments and Suggestions for Authors
Dear Authors,
Thank you for your detailed answers to my comments. You provided detailed responses to all my questions and made appropriate changes in the Revised Manuscript. I only suggest adding the lack of specific tests for sickness behavior in the final paragraph of Disscussion describing limitations of the study.
Kind regards
Author Response
Thank you for your continued time and valuable feedback on our manuscript. We sincerely appreciate the care you have taken in reviewing our work. We have addressed your point, as detailed below.
Comment 1: I only suggest adding the lack of specific tests for sickness behavior in the final paragraph of Disscussion describing limitations of the study.
Response 1: Thank you for bringing this to our attention. We've addressed this oversight by adding this limitation in the final paragraph of Discussion (page 17, lines 619-620):
" It is also important to acknowledge several methodological constraints that shape the interpretation of our results and suggest priorities for subsequent research. Our findings are constrained by the measurement of inflammatory markers at the gene expression level, the lack of temporal tracking of the immune response and the absence of specific tests for sickness behavior."